# Early warning signal for a tipping point suggested by a millennial Atlantic Multidecadal Variability reconstruction

Simon L. L. Michel [1,2] ✉, Didier Swingedouw [2], Pablo Ortega[3], Guillaume Gastineau[4], Juliette Mignot [4], Gerard McCarthy [5] & Myriam Khodri[4]

Atlantic multidecadal variability is a coherent mode of natural climate variability occurring in the North Atlantic Ocean, with strong impacts on human societies and ecosystems worldwide. However, its periodicity and drivers are widely debated due to the short temporal extent of instrumental observations and competing effects of both internal and external climate factors acting on North Atlantic surface temperature variability. Here, we use a paleoclimate database and an advanced statistical framework to generate, evaluate, and compare 312 reconstructions of the Atlantic multidecadal variability over the past millennium, based on different indices and regression methods. From this process, the best reconstruction is obtained with the random forest method, and its robustness is checked using climate model outputs and independent oceanic paleoclimate data. This reconstruction shows that memory in variations of Atlantic multidecadal variability have strongly increased recently—a potential early warning signal for the approach of a North Atlantic tipping point.

Since the early 20th century, the North Atlantic region has exhibited successive decades of anomalously warm and cold sea surface temperatures[1] (SSTs) relative to the global average, effectively amplifying or damping the effects of global warming in the North Atlantic[2,3]. The underlying mode of variability, the Atlantic Multidecadal Variability (AMV), formerly named the Atlantic Multidecadal Oscillation[1], has been linked to a variety of climate effects[4,5], including drought and precipitation in the Sahel[5,6], northeastern Brazil[5], and central Asia[5,6]; hurricane frequency and intensity in the Atlantic[6,7]; sea ice thickness and extent in the Arctic[8]; and climate variability in the Pacific[9]. Therefore, understanding AMV drivers is crucial to accurately predict its future changes and related global and regional impacts.

However, the mechanisms driving the AMV and its dominant timescales of variability remain highly debated. Disagreements stem primarily from the relatively short period of approximately 150 years over which the AMV is directly observed—a time span that includes large anthropogenic climate variations. Early studies considered AMV to be mainly an internal mode of variability, in part because SST variations in the North Atlantic were larger than at a global scale, with a strong variability on multidecadal timescales[4,5,7,10]. However, it has been shown that the last observed period of cold AMV (~1965–2000 C.E.) coincides with both strong anthropogenic aerosol emissions from Europe and North America[11,12] and strong volcanic activity[12], which are both associated with cooling via their radiative impacts. Nevertheless, the assumption that North Atlantic SST (NASST) variations are purely externally driven is challenged by the observed characteristics of the subsurface ocean in the tropical Atlantic, which

[1]Institute for Marine and Atmospheric research Utrecht (IMAU), Department of Physics, Utrecht University, Utrecht, the Netherlands. [2]Environnements et Paléoenvironnements Océaniques et Continentaux (EPOC), Université de Bordeaux, Allée Geoffroy Saint-Hilaire, Pessac 33615, France. [3]Barcelona Supercomputing Center (BSC-CNS), Edificio NEXUS I, Campus Nord UPC, Gran Capitán, 2-4, 08034 Barcelona, Spain. [4]Laboratoire d'Océanographie et du Climat (LOCEAN), Sorbonne université-CNRS-IRD-MNHN, 4 place Jussieu, 75005 Paris, France. [5]Irish Climate Analysis and Research UnitS (ICARUS), Department of Geography, Maynooth University, Maynooth, Ireland. ✉e-mail: s.l.l.michel@uu.nl

were not correctly represented in simulations, including regional aerosol forcing[13].

Internal ocean variability has been identified in coupled Atmosphere–Ocean General Circulation Models (AOGCMs) as a potential key driver of the AMV, mainly through variations in the Atlantic Meridional Overturning Circulation[13–17] (AMOC), a large-scale ocean circulation component transporting warm surface waters northward in the North Atlantic[13]. The complexity and the large three-dimensional spatial extent of the AMOC make it more difficult to measure than surface ocean properties. Direct observations of its strength have only been available since 2004 and at one specific latitude (i.e., 26.5°N)[18,19]. Estimating the potential influence of the AMOC on surface ocean conditions would, therefore, help to understand the AMOC's behavior over larger timescales. In this respect, the link between AMOC and AMV has been highlighted by a wide range of AOGCMs and experiments[13–17]. This connection is often explained by the interhemispheric oceanic heat transport of the AMOC[10], which affects the temperature of the entire North Atlantic. Nevertheless, the dominant timescales of the variability of both AMV and AMOC vary greatly across these simulations[20,21]. This raises legitimate questions about the precise process underlying this mode of variability, as well as how to reliably compare model simulations and observations of the AMV.

In a context where variations in greenhouse gas and aerosol concentrations were smaller than over the historical period, several long paleoclimate records[22,23] and last millennium climate simulations[17] also support the existence of multidecadal variability related to the AMOC during the preindustrial era. Nevertheless, even for these long periods, the role of internal and external factors in producing AMV variations is also debated[21,24].

Although modeling studies and paleoclimate data suggest that AMOC has an effect on the multidecadal NASST variations[13–17,22,23], it is unclear how much the observed NASST is also influenced by external factors. To address this issue, studies have proposed removing external radiative signals when defining the AMV to isolate the intrinsic component of NASST variations driven by AMOC heat transport[2,7,25]. As a result, the current study will concentrate on the latter AMV definition, excluding the direct radiative effects of external forcings by design. To explore the sensitivity of commonly used approaches to remove the radiative effects of external forcings, we will consider three different AMV indices, which are designated as $AMV_T$[2], $AMV_{TS}$[7], and $AMV_F$[25] ("Methods" and Fig. 1a), where the subscripts T, TS, and F refer to associated references[2,7,25]. In contrast to simple detrending methods[25], which can leave spurious signals, the forced variability removed in these AMV indices is expected to account for the evolution of greenhouse gas and aerosol concentrations over the historical period, using different approaches (see "Methods").

The debate on the driving factors of the AMV also leads to uncertainty concerning its spectral properties. The 150-year record of instrumental SST observations does not allow for the identification of consistent frequency bands at multidecadal timescales. Those two limitations (blurred drivers and short timescale) can explain the discrepancies between the AMV periodicity evaluated from instrumental observations[1] (50- to 70-year timescale) and the predominant 10- to 30-year periodicity found in most of the preindustrial AOGCM simulations from the Climate Model Intercomparison Project Phase 5 (CMIP5)[20]. A recent study based on 16 simulations from eight CMIP5 AOGCMs found that transient last millennium simulations including volcanic forcings consistently produce a spectral peak of 50–70 years in global mean surface temperature (GMST), while such a model-wise spectral consistency was not observed for preindustrial control simulations[21]. As a result, they proposed that volcanic eruptions alone could explain this frequency peak in the AMV in historical and paleoclimate data[21]. However, the results from control simulations of the aforementioned study[21] contradict findings from another AOGCM

ensemble study, which used a much larger set of preindustrial control simulations from up to 39 CMIP6 models and found that most of them show a robust relationship between the North Atlantic and GMST variability at interdecadal to multidecadal timescales[26]. Furthermore, an analysis of ice cores and marine archives also calls the hypothesis of a unique role of volcanic eruptions in shaping 50–70-year AMV peaks[19] into question, since it revealed that the 55–70-year AMV timescale was essentially driven by the AMOC and atmospheric variability for the majority of the Holocene[27].

Aside from the fact that the AMV may be strongly linked to AMOC variations[13–17,27], the AMOC is also well-known for its nonlinear dynamics and is considered as potentially unstable[28]. However, the AMOC fate in future climate projections is highly uncertain[29]. Regarding observations, while a recent study based on subsurface densities indicated that there was no discernible trend in AMOC fluctuations over the last 30 years[30], other studies have suggested that its current intensity is abnormally weak when looking over a longer timeframe[31–34]. Indeed, an estimate of the AMOC strength based on SST observations suggests that it may have reached an anomalously weak state these last decades[31], and some paleo-reconstructions even show it to be a record low for at least a thousand years[31–34]. Other recent studies[35,36], however, have revealed that the strength of ocean currents varies widely between regions, ocean depths, and time, casting doubt on the hypothesis of a recent major basin-scale change in the AMOC derived from a relatively small number of oceanic cores[31,33]. In terms of the near-term future, evidence of significant Early Warning Signals (EWSs) for an upcoming tipping point has been recently shown for several indices of the AMOC based on observed surface salinity or temperature of the last ~150 years[37]. However, it is difficult to determine the robustness of these EWSs, especially because of the short timeframe over which they are calculated[38]. In this study, we investigate an AMOC EWS but on a much longer timescale using a reconstruction of one of its surface fingerprints, namely the AMV. In the formalism of dynamical systems theory, EWSs were developed to anticipate critical transitions of a system's internal dynamics as a result of an external perturbation, typically in the form of strong changes in persistence, variance, or skewness[39]. This definition supports our attempt to track a potential approaching tipping point of the AMOC using the AMV definitions depicted above, as it is expected to retrace variability in ocean dynamics rather than external radiative forcings acting on NASST.

Given the lack of SST measurements on sufficiently long timescales, one way to reconstruct the AMV is to rely on its observed climate impacts and fingerprints in specific paleoclimate proxy records. To document these fingerprints, we performed a regression analysis between (i) the mean of the three aforementioned AMV indices (without direct radiative effects) computed from HadISST[40] (back to 1870 C.E.), and (ii) the gridded fields of surface air temperature and precipitation from CRU TS4[41] (Fig. 1b–k). Overall, the AMV fingerprints obtained from this regression analysis are consistent with the findings from previous studies[4–7]. It confirms that the temperature anomalies associated with the AMV are not limited to the North Atlantic region. Indeed, AMV is also associated with temperature changes in southeast Asia, Africa, the Middle East, and western North America (Fig. 1b–k). The AMV fingerprints on precipitation show that its positive phases are associated with more intense summer precipitation over West Africa and drought over western North America and northeastern South America. We also find significant precipitation anomalies over Eurasia (Fig. 1b–k), though with a more scattered and seasonally dependent pattern. These fingerprints are in agreement with modeling studies[9,42] that highlight the widespread and seasonally varying teleconnection patterns of the AMV, including the remote impact over Asia (Fig. 1b–k).

We now focus on AMV variations during the last millennium, a period with a relatively good spatial distribution of high-resolution precipitation and/or temperature-sensitive proxy records, making it a

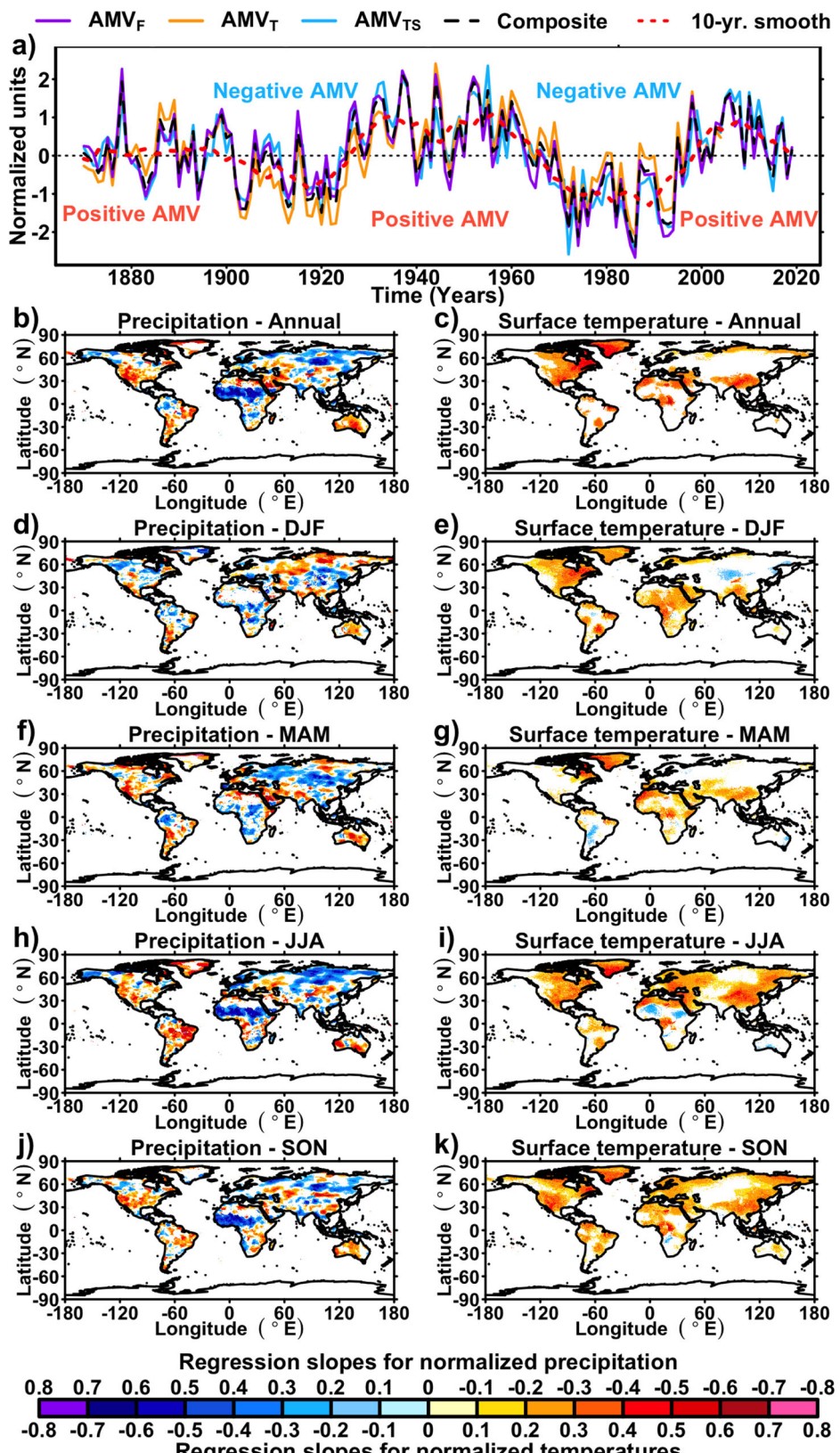

very suitable timeframe for producing paleoclimate reconstructions at annual resolution. Previous attempts to reconstruct the AMV over the last millennium have relied on models with paleoclimate data assimilation[43], paleoclimate data from a single or a few sites[23,27], or statistical regression methods with multiple proxy records as predictors[22,44]. Our study will focus on the latter type of approach

(i.e., regression) to extend back the AMV. In this respect, two previous studies[22,44] reconstructed the AMV over the last millennium using a principal component regression method and continental proxy data located in North Atlantic bordering regions. However, these statistical models were calibrated using raw NASST anomalies[22,44]. As a result, the obtained reconstructions inevitably included the direct radiative effect

**Fig. 1 | Climate impacts of the Atlantic multidecadal Variability (AMV) over the historical era. a** Historical evolution of AMV indices investigated in this study for the period 1870–2017 calculated using the HadISST dataset[40] ("Methods"). **b, d, f, h, j** Maps of averaged regression coefficients between the 10-years smoothed composite of the three AMV indices from panel **a** and CRU TS4[41] precipitation data for the period 1901–2017. Maps are, showing Annual, December–January–February (DJF), March–April–May (MAM), June–July–August (JJA), and September–October–November (SON) regression coefficients, respectively. **c, e, g, i, k** Maps of regression coefficients between the composite of the three AMV indices from panel **a** and CRU TS4[41] surface temperature data for the period 1901–2017. Maps are respectively showing Annual, DJF, MAM, JJA, and SON regression coefficients, respectively. For all maps, white grid points indicate that regression coefficients are not significantly different than 0 at the 90% confidence level, using a two-tailed student test with corrected degrees of freedom[45] ("Methods").

of external forcings[44], and thus does not correspond to the AMV as defined in the present study[2,7,25]. In addition, former reconstruction studies[22,44] do not include any statistical tests for the selection of the proxy records, which means that some of them may not be proper predictors of the AMV. The present study thus relies on statistically objective methods in the selection of proxy data[45] from the PAGES 2k database[46] ("Methods").

In contrast with previous studies that focused on a single reconstruction, we here use an ensemble of 312 reconstructions to investigate the sensitivity to (i) the target observed AMV index, (ii) the regression method, and (iii) the calibration period over which the statistical models are constructed[45]. Each of these reconstructions is evaluated with appropriate validation scores[47] ("Methods"), which are then used to identify the best-performing one that is selected as the final reconstruction. This best-performing reconstruction is subsequently validated using climate model outputs[48] and independent ocean records[46]. We later present a spectral analysis of its variability and explore its potential modulation by radiative effects of natural external forcings (solar and volcanic) over the last millennium. Finally, we show a significant EWS in our AMV reconstruction, indicative of the approach to a climate tipping point, and we use millennial-scale climate simulations to illustrate that the EWS metric applied to the AMV can be a relevant indicator for detecting an EWS in the AMOC.

## Results

### Optimal selection of the regression model and validation

The present study compares 312 reconstructions produced by combining 4 regression methods (principal component regression, random forest, partial least squares, and elastic-net, Supplementary Note 1), 3 definitions of the AMV index (AMV$_T$[2], AMV$_{TS}$[7], AMV$_F$[25], "Methods"), and 26 different time windows over which the regression model is constructed (Methods). The reconstructions are performed with the ClimIndRec[45] toolbox, which evaluates their performance with Coefficient of Efficiency scores ($S_{CE}$) and optimizes the specific control parameters of each regression method by k-fold cross-validation[45,49] (see "Methods").

Each reconstruction covers the preindustrial period up to 850 C.E. and is made from a selection of proxy records significantly correlated (at the 95% confidence level) with the observed AMV index. This selection is realized separately for the three AMV definitions and timeframes investigated ("Methods"). The year 850 C.E. was chosen since it is the starting point for the last millennium climate simulations. Such simulations will be used later to further evaluate and validate our reconstruction in a pseudo-proxy framework[48]. The observed time series for the three AMV indices all reach back to 1870 C.E., the first year for which gridded SST observations are available[40]. The selection of the proxy records is made from a large set of annually resolved proxies from the Northern Hemisphere, which includes the PAGES 2k database[46] and 41 others[45] (Supplementary Table 1). This database (hereafter P2k+) consists of 457 records and was constructed using different quality criteria (Methods). An important preprocessing step is to remove from each P2k+ proxy record an estimate of the forced variability of NASST ("Methods"). This approach is one of the main differences with respect to previous NASST/AMV reconstructions[22,44] which mixed forced and internal variability signals. Here, to account only for the internal AMV signature in the proxy records, the component of the radiative effects from external forcings is first estimated

using a signal-to-noise maximizing empirical orthogonal function technique[2] ("Methods" and Supplementary Fig. 1) based on NASST time series from the average of historical simulations of 32 CMIP5 models (Supplementary Table 2). It should be noted that the forcing component estimated from CMIP5 historical simulations is preferred here over that estimated from CMIP6 models because a recent study found that CMIP6 historical simulations may overestimate the North Atlantic's response to changes in aerosol concentrations[50]. Nevertheless, the inclusion of small to moderate eruptions in CMIP6 forcings largely influences the GMST response compared to CMIP5 simulations and might have implications for simulated climate variability[51]. As a result, we tested the robustness of our results to the uncertainty in the CMIP generation used by reproducing the various reconstructions and analyses of this study, but estimating the NASST forced component from the CMIP6 historical simulations instead (Supplementary Table 3 and Supplementary Figs. 2–5).

To evaluate and compare each of the 312 reconstructions produced, we compute $S_{CE}$ scores[48] over 30 pairs of randomly drawn training/testing periods, each of which is a partition of the learning period under consideration[45] (i.e., the entire period over which the statistical model is evaluated and constructed, "Methods"). This $S_{CE}$ metric quantifies the predictive ability of the reconstruction methods over the testing periods[45,47]. When the $S_{CE}$ is significantly positive for a given testing sample, it indicates that the statistical model provides better estimates than the empirical mean of the AMV observations from the testing sample[45,47] ("Methods"). Among the 312 reconstructions, the random forest[52] (RF) method provides the reconstruction with the highest mean $S_{CE}$ score, although it is not necessarily the best method when considering the average of the scores by method for all the reconstructions produced (Supplementary Fig. 6). The best mean $S_{CE}$ score calculated over testing periods is obtained when RF is applied to the AMV$_F$ index, for the 850–1987 reconstruction period (i.e., with the 1870–1987 learning period). For this specific reconstruction, the average $S_{CE}$ score over 30 training/testing splits of the learning period is positive at the 99% confidence level, validating its use for reconstruction purposes[47] (med(CE) = 0.25, mean(CE) = 0.23, "Methods").

A shortcoming of the four regression methods considered in this study is that they do not permit the use of paleoclimate records with a temporal gap over either the learning or reconstruction periods. To overcome this limitation, the reconstruction setup associated with the highest $S_{CE}$ scores for the period up to 850 C.E. was subsequently applied in a nested reconstruction framework[44]. This entails gradually reconstructing the AMV$_F$ index over time (1 year at a time, starting with the oldest) by constructing a new RF model that includes newly available proxies at each timestep[44] ("Methods"). This nested reconstruction procedure stops in 1869 C.E., the year before the observed AMV indices start.

To gain more confidence in the trustworthiness of this final reconstruction, we have used, in addition to the standard evaluation metrics within the reconstruction framework[45] ("Methods"), three independent validation approaches:

1. A benchmark nested reconstruction was produced using randomly generated red noise processes as predictors, with similar spectral characteristics than the actual proxy records[53]. This reconstruction was made with the same number of predictors, target AMV index (AMV$_F$), regression method (RF), and

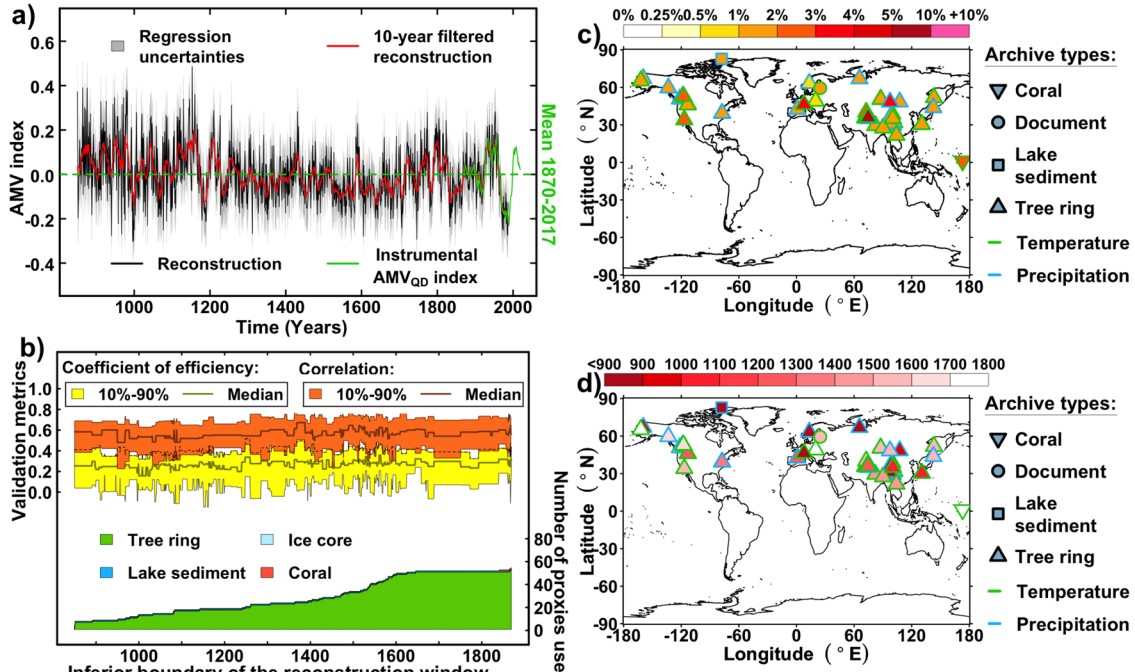

**Fig. 2 | Nested reconstruction of the Atlantic Multidecadal Variability (AMV) and related proxies. a** Black line: Annually resolved nested reconstruction of the AMV_F index using random forest[52] (see "Methods"). Red line: 10-years kernel smoothed nested reconstruction, black line: annually resolved nested reconstruction. The regression uncertainties of the annually resolved nested reconstruction (black line) are defined for each timestep of the nested reconstruction as ±2 standard error of the regression. Green line is the time series of the instrumental calculated from historical SST data[40]. **b** Validation metrics (coefficient of efficiency in yellow and correlation in orange) obtained for 30 training-testing splits, and proxy records' types and availabilities for the nested reconstruction (bottom). **c** Proxies weights from the random forest method, relative to the proxy records temporal availability ("Methods"). **d** Temporal coverage of the availability of the proxy records.

reconstruction period (850–1987) as the final reconstruction (see Supplementary Fig. 7 and "Methods" for further details).

2. A comparison with independent ocean proxy records from the Ocean2k database[46] with multi-annual to multidecadal resolution, in order to assess whether our reconstruction agrees with those slowly varying oceanic changes (see Supplementary Fig. 8, "Methods" and Supplementary Note 2 for further details).

3. Two pseudo-proxy experiments based on climate model outputs[48] to assess (i) the robustness and reliability of the reconstruction method, and (ii) the potential effect of nonstationarities in the connection between the proxies and the AMV (see Supplementary Figs. 9 and 10, "Methods", and Supplementary Note 3 for further details).

In addition, we independently performed a NASST reconstruction where no radiative forcing effect is removed from the predictand (NASST) and predictors (proxy records) with the same optimal model selection (Methods, Supplementary Fig. 11). The purpose of this additional reconstruction is to check whether our reconstruction procedure for the AMV consistently produces no direct response to external radiative forcings.

### Selected proxy records for the best reconstruction

The nested RF reconstruction of the AMV uses a total of 55 proxy records from the Northern Hemisphere (Fig. 2a, b) selected based on their correlation with the AMV_F index. Their final weights, given by RF variables' importance ("Methods"), and their temporal availability, are shown in Fig. 2c, d. We identify three main groups of records with similarly distributed weights: Central Asia, Europe, and Western North America. The relatively low number of European proxies contrasts with previous regression-based reconstructions that preselected proxies solely on the basis of their geographic proximity to the North

Atlantic[22,44]. The fact that only five proxy records from Europe are used could be due to their reduced presence in the proxy record database we consider (<10% of the total of P2k + ), or could simply indicate that the climate signal and seasonality of most European proxies are not representative of the AMV signatures (Fig. 1). Among the five selected European proxies, one of them has a large weight (>3.5%) and interestingly covers the whole reconstruction period (Fig. 2c, d). It corresponds to a time series of tree ring growth measurements in the European Alps[54], which is strongly correlated with summer instrumental temperature over the historical period ($r = 0.7$, $P < 0.01$, Supplementary Table 4). The other four European proxy records are related to summer or annual temperature and precipitation, and are thus in agreement with the well-documented fingerprints of the AMV on European summer temperatures[4,5], also highlighted here in Fig. 1. The proxy data from Asia and western North America are strongly represented in the PAGES 2k[46] database, which may partly explain their relatively large presence in the reconstruction. The selected proxy records from western North America are most sensitive to summer and annual variations in temperature and precipitation (Fig. 2c, d and Supplementary Table 4), although the correlation remains moderate over the instrumental period in this region (Fig. 1). Interestingly, the reconstruction also selects several strongly weighted proxy records of annual and boreal summer (June–July–August) temperature over eastern Pakistan and the Tibetan Plateau. This link is supported in some recent studies that have highlighted the role of AMV variations for spring and summer temperatures in this region, through the impact of the AMV on large-scale atmospheric pressure gradients in the Eurasian sector[55,56]. The reconstruction further includes a large number of proxy records from East Asia/North China, for which the climate conditions have also been shown to be significantly affected by AMV variations through atmospheric Rossby waves propagation and heat advection changes in the western Pacific[57]. Several studies showed

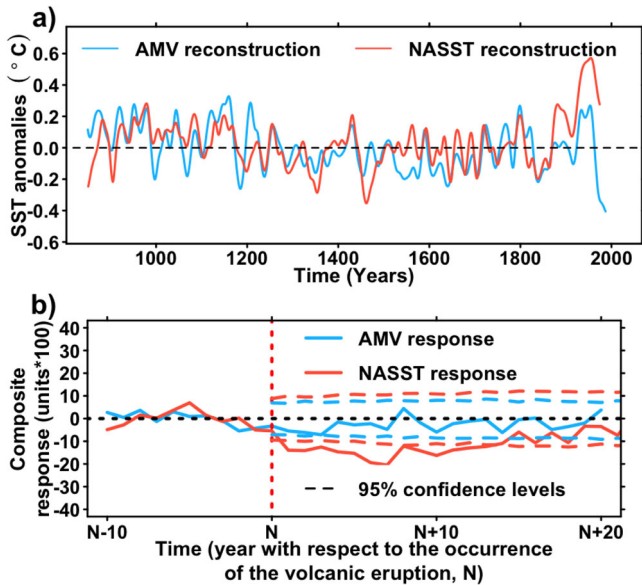

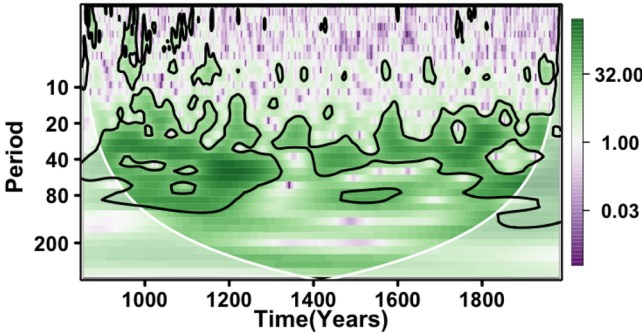

**Fig. 4 | Discrete wavelet transform of the nested Atlantic Multidecadal Variability reconstruction from this study.** Contours provide the 90% confidence level of significance. The white line and the light white-shaded area below indicate the cone of influence. The cone of influence gives the spectrum borders where the edge effect (i.e., the time boundary effect) becomes too important, which cannot be robustly interpreted.

**Fig. 3 | Comparison of the Atlantic Multidecadal Variability (AMV) with North Atlantic Sea Surface Temperature (NASST) and volcanic forcing. a** Final reconstructions of AMV and NASST in sea surface temperature anomalies (°C). **b** Superposed epoch analysis[45] for responses of the AMV and NASST reconstructions to the ten largest eruptions[58] of the last millennium (see Supplementary Table 5). Composite series are performed for 31 years, with the 11th year being the year of the eruptions. Each individual response is centered to its values 10 years before the eruption (from N-10 to N-1, where N describes the year of occurrence of the eruption) before computing the composite time series. 95% confidence levels have been calculated using a Monte-Carlo approach[45].

that the tropical Pacific and the AMV are closely linked through the propagation of large-scale equatorial waves and air–sea interactions, also impacting climate in eastern Asia[9,57]. Thus, although previous AMV reconstructions did not consider proxy records from Asia[22,44], we argue that there are good physical and objective reasons for their selection here[9,55–57].

A detailed description of each proxy record used for our reconstruction and their correlations with the instrumental data and AMV is given in Supplementary Table 4. Their locations are shown on top of the maps from Fig. 1 in Supplementary Fig. 12.

### Response to external forcing of the reconstructions

We now investigate the response of the AMV to relevant natural external forcings before the industrial era, namely aerosol emissions from volcanic eruptions and changes in solar irradiance. Because by construction, the AMV was separated from the direct radiative effect of the external forcings[25], the analysis focuses on responses that are either nonlinear or delayed. The analysis is complemented with a characterization of the fully forced responses of NASST, for which reconstruction has been performed in the same way (i.e., nested approach and methodological choices) as the AMV one but without removing any forced effect from the predictand (NASST) and predictors (proxy records). The best NASST reconstruction produced this way correlates significantly with the AMV reconstruction ($r = 0.64$; $P < 0.01$, Fig. 3a, "Methods" and Supplementary Fig. 11).

Using a recent reconstruction of volcanic activity[58], we perform a superposed epoch analysis[45] (Methods) on the NASST and AMV reconstructions to characterize their radiative response to the 10 largest eruptions of the last millennium (Supplementary Table 5). While the NASST reconstruction shows a decadal-long response after the eruption, similar to the result from a previous study[44], no significant response is found for the AMV reconstruction (Fig. 3b). The volcanic forcing might affect the AMV with a time lag, at multidecadal

timescales, through internal variability excitation and dynamical oceanic response[59]. However, such a lagged response is not detected in Fig. 3b. Individual responses of the AMV to each volcanic eruption are presented in Supplementary Fig. 13.

Concerning solar forcing, neither the 10-year filtered time series nor the 30-year filtered time series of the total solar irradiance (TSI) reconstruction used in the PMIP3 protocol[60] is significantly correlated with our 10-year filtered AMV reconstructions, even when solar forcing leads by a few years ($r = 0.23$, $P > 0.2$, lag = 12; $r = 0.32$, $P > 0.2$, lag = 13; respectively, where $r$ stands for the maximal cross-correlation across possible lags, expressed in years, cf. Supplementary Fig. 14). On the other hand, the 10-year and 30-year filtered time series of the TSI reconstruction do show a modest correlation with the NASST reconstruction when the TSI leads by 13–14 years ($r = 0.5$–$0.52$, $0.1 < P < 0.2$).

Our results thus suggest that natural external forcing does not have a direct influence on the phasing of the AMV, while, as expected, it could play a direct role on variation of the NASST reconstruction. Therefore, the volcanic forcing is not found to act as a pacemaker for the AMV, at least for the ten largest volcanic eruptions of the last millennium.

### Spectral analysis

The wavelet analysis in Fig. 4 shows that the AMV reconstruction exhibits significant multidecadal variations. It varies primarily in the 20- to 90-year band, except for the 1400–1800 period, which is dominated by shorter cycles (20- to 40-year band). Thus, the 50- to 70-year preferential periodicity suggested by observations since 1870[1] may not be systematic. The wide range of AMV preferential timescales can also be found in control simulations of climate models[26,61] and was also obtained in an AMV reconstruction based on data assimilation[43].

To investigate the robustness of these results, the spectral analysis was also extended to the best 30 reconstructions in terms of $S_{CE}$ scores (Supplementary Table 6). We notice that none of these 30 reconstructions' spectral characteristics suggest a frequency range as narrow as the 50- to 70-year timescales, as suggested by the short instrumental period[1] (Supplementary Fig. 15). Instead, all reconstructions also show a dominant variability in the lower part of the multi-decade range (e.g., 20–50 years).

### An early warning signal for an approaching tipping point

It has been shown that detecting the approach of an AMOC regime shift in climate models may require knowledge of hundreds of years of AMOC variations[38], which is far from what direct observations at 26.5°N, currently covering less than 20 years[18,19], provide. In this

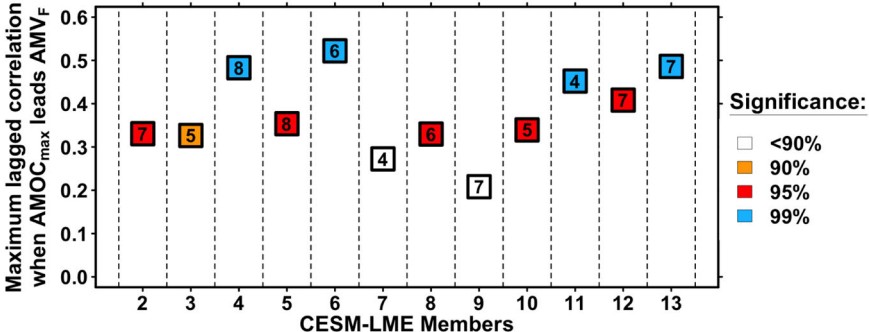

**Fig. 5 | Maximum lagged correlations when the Atlantic Meridional Overturning Circulation index (AMOC$_{max}$) leads the Atlantic Multidecadal Variability index (AMV$_F$) index in 12 members of Community Earth System Model Last Millennium Ensemble (CESM-LME).** Squares correspond to the maximum lagged correlation when AMOC$_{max}$ leads the AMV$_F$ index. Orange, red, and blue colors indicate 90%, 95%, and 99% confidence levels, respectively. White points indicate no significance at the 90% confidence level. Numbers in each point indicate the timestep where the maximum cross-correlation is reached. All time series are smoothed with a 10-year kernel filter. Exact cross-correlation functions for each member are given in Supplementary Fig. 18.

regard, the evaluation of ocean surface indices as reliable estimates of AMOC variability is debated in the climate community. Indeed, while the heat transport associated with the AMOC can certainly contribute to changes in surface ocean properties[10,13–17], NASST also responds strongly to the direct radiative effect of external forcings[11,12,21,24,44] (Fig. 3b). However, our reconstruction is, by construction, expected to be largely controlled by the dynamics of North Atlantic ocean circulation, and notably of the AMOC[10,13–17,27]. Indeed, due to its deep ocean aspect, the AMOC may not respond to direct variations in radiative forcings, which is consistent with our AMV reconstruction (Fig. 3b and Supplementary Fig. 14) that we expect to be suitable to retrace AMOC variations. However, delayed and nonlinear dynamical response of the AMOC to other kinds of externally forced perturbations, such as freshwater input in the northern North Atlantic, may still be detected since it is not being removed in AMV indices.

To verify the linkages between the AMOC and AMV, their simulated indices are first investigated in 12 members of CESM-LME[62]. To date, the CESM climate model is the only one that provides such a large number of simulations for the last millennium, making it an invaluable framework for studying this period[62]. Its use here is particularly relevant because historical runs are too short to study multidecadal relationships between time series. In addition to the simulated AMV$_F$ indices for the last millennium extracted from CESM-LME for the pseudo-proxy experiment ("Methods" and Supplementary Note 3), we also calculated the corresponding AMOC indices. To quantify the AMOC strength in simulations, we used the maximum stream function value below 500 meters and between 20°N and 65°N (AMOC$_{max}$). Figure 5 shows, for each CESM-LME member, the maximum lagged correlations when the AMOC$_{max}$ index leads the AMV$_F$ (using decadally smoothed time series). The associated cross-correlation functions are presented in Supplementary Fig. 16. We find that 10 out of 12 CESM-LME members show maximum and significant correlations at the 90% confidence level when the corresponding AMOC$_{max}$ leads by 4–8 years. Among these members, the confidence level reaches 95% for 9 of the members and 99% for 4 of them. Similar percentages and lead times are obtained when other alternative AMOC indices are considered (Supplementary Figs. 16 and 17). The discrepancies observed among members in terms of significance level demonstrate the relevance of using an ensemble of simulations to properly study the relationships between climate phenomena, especially for such a long timescale and when using a single AOGCM[62]. These discrepancies may reflect the different internal variabilities sampled among the ensemble members[62].

We test whether an AMV reconstruction can detect a potential AMOC tipping point, using the relationships between the AMV and the AMOC found in CESM-LME (Fig. 5) and various other model simulations in previous studies[10,13–17]. We employ a tipping point detection metric, the critical slowing down EWS—a type of EWS which has previously been used in studies based on simulated AMOC[38,39] or observed AMOC indices based on surface observations[37]. Dynamically, this approach assumes that a given system might be approaching a bifurcation if its memory grows with time, i.e., the state of the system at time $t+1$ becomes increasingly dependent on the state of the system at time $t$. In other words, as the system's memory grows, so does the time it takes to recover from variations. In time series analysis, an increasing memory of the system is reflected by an increasing first-order autoregressive (i.e., AR(1)) coefficient[39]. Kendall $\tau$ statistics are calculated for the AR(1) coefficients to estimate their evolution in time for different sliding window lengths[38,39] (WL, WL = 200–400 years with a 50-year increment) in order to evaluate such an EWS in the AMV reconstruction. Here, the Kendall $\tau$ statistic quantifies the AR(1) coefficients' temporal evolution as the ranked correlation between the AMV's sliding AR(1) coefficients and time ("Methods"). Although this EWS approach has been shown to be successful in an earth system model of intermediate complexity[38,39], there are still some sources of uncertainties about its application, such as the lack of information on the exact time when the tipping point is crossed, as well as the Kendall $\tau$ statistic's dependence on the time span it is calculated on.

This EWS test applied to our "real" paleoclimate AMV reconstruction indicates a highly significant increase in the AMV's memory over the recent period for all window lengths: $\tau \in [0.47, 0.55]$ ($P < 0.01$ for all, Fig. 6a, "Methods"). According to the theory of tipping point detection in climate time series[39], this is a robust estimate, as it is based on sufficiently long observations[38], that the AMV may now be approaching a tipping point after which the Atlantic current system might undergo a critical transition. Figure 6a also shows that the most recent values of the AR(1) have seen a particularly sharp increase. Indeed, when only the last 200 years of the time-varying AR(1) coefficients are considered, they reach much higher levels of increase: $\tau \in [0.86, 0.94]$ ($P < 0.01$ for all WL, Fig. 6a, "Methods"). The top 30 reconstructions (Supplementary Table 6) also suggest a significant EWS, regardless of the length of the WL considered ($\tau \in [0.39, 0.77]$ for all 90 tests, $P < 0.01$ for 89 of the 90 tests, Supplementary Table 7 and Supplementary Fig. 18), showing that this could not be found by chance on the single reconstruction with the highest $S_{CE}$ scores.

To determine whether the AMV-based EWS can be interpreted as an EWS for the AMOC, we also provide a similar analysis performed on those CESM-LME simulations for which a significant lagged AMOC/ AMV relationship was identified in Fig. 5. We computed EWS AR(1) statistics from the CESM-LME simulations for the AMOC and the AMV$_F$ indices, for three different window lengths[39] (WL = 200, 300, and 400).

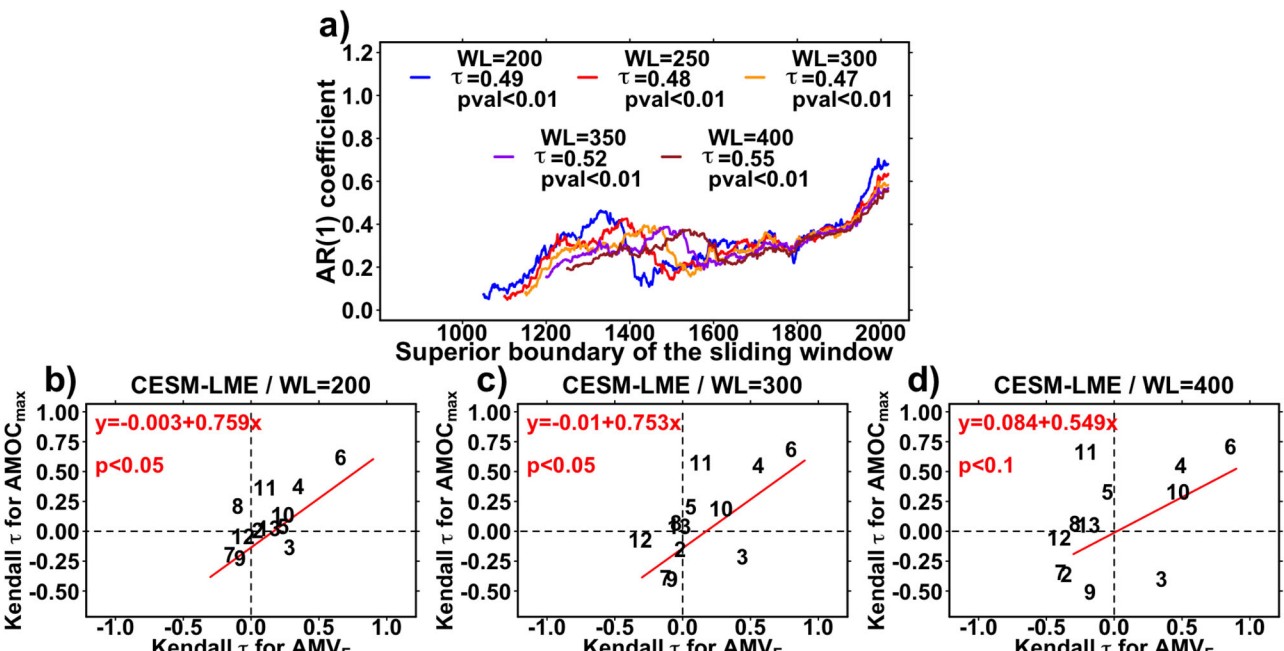

**Fig. 6 | Early Warning Signal (EWS) test of the Atlantic Multidecadal Variability reconstructed index (AMV$_F$) and relevance in Community Earth System Model Last Millennium Ensemble (CESM-LME) simulations. a** EWS for the reconstruction. The applied test is based on first-order autoregressive coefficients (AR(1)), for different window lengths (WL)[38, 39]. For each WL, sliding AR(1) coefficient are computed, and a Kendall τ statistics between time and the sliding AR(1) time series are calculated (see "Methods"). Significances are approximated using Gaussian distributions because of the large length (>50) of the AR(1) coefficients (see "Methods"). **b–d** EWS statistics in CESM-LME simulations. Kendall τ statistics obtained for the maximum Atlantic Meridional Overturning Circulation strength below 500 meters depth (AMOC$_{max}$) AR(1) coefficients (ordinates) are plotted against the Kendall τ statistics obtained for the AMV$_F$ AR(1) coefficients (abscissas) for WL = 200 (**b**), WL = 300 (**c**), and WL = 400 (**d**), respectively. Plotted numbers indicate the member index of the CESM-LME simulations (from 2 to 13). Red lines are the ordinary least squares regression lines, and their significance is calculated using a two-tailed Student t test of the regression slope.

The Kendall τ statistics obtained for the AMV$_F$ index are, for the most part, consistent with those obtained for the AMOC$_{max}$ index (Fig. 6b–d). According to the linear regressions between the AMOC$_{max}$ and AMV$_F$ statistics (Fig. 6b–d), AMV may slightly overestimate the AMOC autoregressive properties, but it accurately captures these changes on a millennial-scale (at least $P < 0.1$ for WL = 200 to 400, Fig. 6b–d). Using these linear regression models computed in CESM-LME between the AMV$_F$ and AMOC$_{max}$ indices (Fig. 6b–d) and the EWS statistics calculated from the AMV reconstruction (Fig. 6a), we estimate the Kendall τ of AR(1) coefficients from the last millennium AMOC to be 0.37, 0.34, and 0.39 for WL = 200,300,400, respectively ($P < 0.01$ for the three; see "Methods").

## Discussion

The AMV reconstruction proposed here is based on robust optimization and model selection techniques for statistical regression[45,47,49]. This AMV reconstruction of the last millennium is based on a total of 55 proxy records[45,46] (Fig. 2) and has been validated against a reconstruction based on surrogate time series with the same persistence as the proxy records[53] (Supplementary Fig. 7), AOGCM simulation outputs of the last millennium[62] (Supplementary Figs. 9 and 10), and independent ocean proxy records[46] (Supplementary Fig. 8).

This AMV reconstruction provides an indirect estimate for past variability in the ocean circulation and notably the AMOC, as verified in last millennium model simulations[17] (Figs. 5 and 6b–d). As previously stated, the AMV in this study is reconstructed in such a way that the direct radiative influence of external forcings is not included. However, delayed effects of the forcings on the AMV, in terms of intensity, persistence, or changes in spectral characteristics, may remain as a result of its interactions with other climate components responding to forcings. On top of the absence of the direct radiative effect of natural forcings, the AMV reconstruction varies in a wide frequency band

(20–90 years, Fig. 4), which is larger than the variability estimated at 50–70 years from observations[1], but agrees with data assimilation[43] and modeling[26,61] studies.

We also found a significant EWS for an approaching tipping point based on this AMV reconstruction (Fig. 6a). Tipping occurs as a critical threshold, or tipping point, is crossed, leading the system into a different dynamical regime[39]. In this context, while our AMV reconstruction does not show a consistent radiative response to natural forcings (Fig. 3 and Supplementary Figs. 13 and 14), it may primarily reflect changes in the AMOC internal variability caused by the forcings (Figs. 5 and 6b–d). It must be emphasized that the critical slowing down EWS we employ may generate false alarms for a variety of dynamical systems[63,64]. Nevertheless, in contrast to the case of the AMV/AMOC with, e.g., underlying and diffuse anthropogenic greenhouse gas forcing causing strong and persistent freshwater anomalies in the northern North Atlantic, false alarms may especially occur for systems that are not prone to tipping[63]. Therefore, based on the relationship found in models between the AMV and AMOC at decadal timescales[13–17] (Figs. 5 and 6b–d), relating the EWS found in the AMV with potential large upcoming changes in the AMOC, a well-known tipping element of the climate system[28], appears to be a reasonable and consistent assumption.

Future observations of the actual AMOC strength are still required to draw a reliable picture of its exact impact on the AMV, as this assumption is still primarily based on AOGCM simulations[13–17] (Figs. 5 and 6b–d). As a result, more observation-based evidence of the precise AMOC impacts on ocean surface properties remains crucial in understanding the extent to which EWSs observed at the ocean surface[37] (Fig. 6a) can provide robust insights into future AMOC variations. Consequently, the discovered AMV EWS could be a sign of an approaching bifurcation of a more localized feature of the AMOC, such as an abrupt convection weakening in the North Atlantic subpolar gyre

region, as observed in some future projections from both the CMIP5[65] and CMIP6[66] AOGCM ensembles.

The debate over the fate of the AMOC has intensified recently[30,36], as evidenced by the fact that many paleoclimate records are inconsistent with the current strength of the AMOC being below normal[35,36], and subsurface density observations extending back 30 years show no consistent trend in the AMOC strength over this period[30]. A study[67] recently critiqued these various statements, arguing that some paleoclimate proxies that do not show an AMOC decline[35] may not necessarily reflect its variations. Moreover, this study[67] illustrated that the absence of a trend in the AMOC over the last 30 years[30] does not contradict its longer estimates based on a paleoclimate reconstruction[32] and observations[31] of SST fields. The growing evidence from various sources that a strong AMOC state change might be on-going[31–34,37] must be taken seriously in the panel of plausible scenarios in order to better communicate the potential consequences this implies and thus better support decision making[68]. Indeed, such a well-documented AMOC state bifurcation could have significant climate and ecological consequences[28,69,70]. Therefore, this finding emphasizes the importance of taking into account the potential consequences of such a large change in the Atlantic in climate adaptation plans[68].

## Methods

### Estimation of NASSTs forced component using signal-to-noise maximizing EOF

The forced component of NASST (all North Atlantic grid points between latitudes 0° and 60°N) is estimated from historical simulations of 32 coupled climate models (Supplementary Table 2). NASST time series from the 32 simulations are extracted and merged as columns of the same matrix. Using a principal component analysis of the latter matrix, the first principal component is retained as the estimated forced component of NASST (Supplementary Fig. 1).

### Instrumental AMV indices

Historical AMV indices have been calculated using annually resolved observations of SST data from the Hadley Center dataset (HadISST) for the period 1870–2019. Three instrumental AMV indices are defined as the spatially averaged SST over the North Atlantic region, differing in the way the externally forced signal is removed, either beforehand or afterward.

The first index (AMV$_{TS}$, Fig. 1a) uses the global averaged SST anomalies between 60°S and 60°N as a proxy for the externally forced signal that is subtracted from NASST.

A second index (AMV$_T$; Fig. 1a), is built using climate model historical simulations to isolate the forced component in NASST. It is calculated with a signal-to-noise maximizing empirical orthogonal function ("Methods") that is then removed by linear regression from an estimate of its 10-year smoothed effect at each grid point of the North Atlantic. The AMV$_T$ index is then obtained as the average of regression residuals over the North Atlantic region.

Finally, the AMV$_F$ is obtained as the spatial average of the North Atlantic time series residuals obtained after regressing out the ten-year moving average of the global mean SST between 60°S and 60°N.

Only after being reconstructed, each AMV index has been smoothed with a 10-year kernel filter, following the classical AMV definition.

### Proxy records database

To select the proxy records of this study, a first large dataset is made by merging the PAGES 2k database (686 records) with other proxy records in neighboring continents of the North Atlantic published by different studies (Supplementary Table 1). Duplicates from the three sources are removed leading to a final database of 727 proxy records (P2k-ALL). The proxy records from P2k-ALL used in the reconstruction have been selected to fulfill the following conditions: (i) they are

annually resolved, (ii) they are located in the Northern hemisphere (latitude >0°), and (iii) they are significantly correlated at the 95% confidence level with at least one historical time series of either annual or seasonal precipitation or surface temperatures from the nearest grid point within the CRU TS4 historical dataset.

First analyses singled out a proxy record in Asia named "Asia.MOR1JU" which had abnormally large RF weights (more than 5 times higher than the second) such that $S_{CE}$ scores were almost doubled due to its inclusion. The reason for this behavior was not clear but highlighted a limitation of the nonlinear method. Whatsoever, since hundreds of proxy records are tested against the AMV indices, such a statistical anomaly does not appear unlikely. In order to prevent biasing the reconstruction toward this single proxy record, we decided to remove it from the database used in this study.

In the next sections, we describe how different reconstructions are compared and how a final nested reconstruction of the AMV is obtained. These reconstructions also use correlation tests to select proxies from P2k+ that are significantly correlated with a given AMV index, for a given learning and reconstruction period. This means that only a subset of the most relevant proxy records from the P2k+ database is finally used in each reconstruction.

### Generation of the AMV and NASST reconstruction ensembles

The 312 reconstructions compared in this study are performed for 26-time windows $\Gamma$: from 850–1975 to 850–2000, subsequently incrementing the superior boundary by 1 year. We use this approach to sample the sensitivity of the reconstructions to the calibration period, which goes from 1870 to the upper bound of $\Gamma$. For lower upper bounds, more proxies are available, but the regression models are built over shorter calibration periods and therefore have fewer degrees of freedom. These 26 temporal windows are used in combination with three AMV indices and four regression methods. A detailed description of these regression methods is given in Supplementary Note 1. We thus obtain $26 \times 3 \times 4 = 312$ final reconstructions which are compared using the $S_{CE}$ metric which allows us both to determine the best model and to say how reliable it is (next Methods section, Supplementary Fig. 6). All these setups are tested by only using proxies available and significantly correlated at the 95% confidence level with the respective AMV index. Noteworthy, it is, of course, possible that some proxies correlate significantly with AMV just by chance, which implies that some spurious predictors might be used in the reconstruction. However, the weights of these proxies are expected to be low, as they should be penalized in the cross-validation process.

For the NASST reconstruction, 104 setups are compared by shuffling the same 26 temporal windows and the same 4 regression methods as for each of the AMV indices. As for the AMV, the best-performing NASST reconstruction is retained as the one with the highest $S_{CE}$ scores ("Computation of reconstruction and evaluation" and Supplementary Fig. 11).

### Computation of reconstruction and evaluation

We define the reconstruction period as $\Gamma$, defined by $N$ annual time steps, and the common period of the proxy records and the AMV index as $\mathbf{T}$, in this case defined by $n < N$ annual time steps such that $\mathbf{T} \subset \Gamma$. We then define the AMV index as $\mathbf{Y} \in \mathbb{R}^n$ and the matrix of the $p$ available proxy records as $\mathbf{X} \in \mathbb{R}^{N \times p}$. We finally denote as $\mathbf{x} \in \mathbb{R}^{n \times p}$ the submatrix of $X$ that contains the proxy records values over the timeframe $\mathbf{T}$. The $\mathbf{X}$ matrix can then be denoted as $\mathbf{X} = [(\mathbf{X}_t^j)_{t \in \Gamma}]_{1 \leq j \leq p}$ and $\mathbf{x} = [\mathbf{x}^j]_{1 \leq j \leq p} = [(\mathbf{X}_t^j)_{t \in \mathbf{T}}]_{1 \leq j \leq p}$.

We then randomly split $\mathbf{T}$ in $R = 30$ pairs of training/testing samples respectively denoted, $\forall 1 \leq r \leq R$, as $\{\mathbf{x}_{(train)}^{(r)}; \mathbf{Y}_{(train)}^{(r)}\}$ and $\{\mathbf{x}_{(test)}^{(r)}; \mathbf{Y}_{(test)}^{(r)}\}$. Here, the training sample size is set to be 80% of the length of $\mathbf{T}$ so, by extension, the testing sample size is 20% of the length of $\mathbf{T}$.

 

For statistical modeling, we use $\mathbf{Y}^{(r)}_{(train)}$ as the predictand and $\mathbf{x}^{(r)}_{(train)}$ as the predictors. For a given regression method denoted M, we apply KFCV (see subsection "k-fold cross-validation (KFCV)") to each training set $\left\{\mathbf{x}^{(r)}_{(train)}; \mathbf{Y}^{(r)}_{(train)}\right\}$ as a metric to find the optimal set of parameters associated to the training sample and M.

M and the associated optimal set of control parameters are then applied to $\mathbf{X}^{(r)}$ in order to reconstruct $\mathbf{Y}^{(r)}$ on the testing period, which gives $\hat{\mathbf{Y}}^{(r)}_{(test)}$, and the reconstruction period, which gives $\hat{\mathbf{Y}}^{(r)}_{(rec)}$. This requires that $\hat{\mathbf{Y}}^{(r)}_{(test)} = (\hat{\mathbf{Y}}^{(r)}_{(rec)})_{t \in \mathbf{T}}$. The validation score associated to the $r^{th}$ training sample is then calculated using the $S_{CE}$[47] over the $r^{th}$ testing sample:

$$s^{(r)} = S_{CE}\left(\hat{\mathbf{Y}}^{(r)}_{(test)}, \mathbf{Y}^{(r)}_{(test)}\right) = 1 - \frac{\sum_{i=1}^{m}\left(\mathbf{Y}^{(r)}_{i(test)} - \hat{\mathbf{Y}}^{(r)}_{i(test)}\right)^2}{\sum_{i=1}^{m}\left(\mathbf{Y}^{(r)}_{i(test)} - \bar{\mathbf{Y}}^{(r)}_{(test)}\right)^2}, \text{ with } \bar{\mathbf{Y}}^{(r)}_{(test)} = \frac{1}{m}\sum_{i=1}^{m}\mathbf{Y}^{(r)}_{i(test)} \quad (1)$$

Where $m$ is the length of the testing sample.

This validation score gives an estimation of the accuracy of the statistical model when reconstructing the observed variability not included in the reconstruction period. When $S_{CE} < 0$, it means that a simple sample average over the testing period is better than the output given by the statistical model. Contrarily, $S_{CE} > 0$ means that the statistical model gives a more reliable reconstruction than the average over the testing sample[45], the associated reconstruction is thereby considered as reliable in this study.

The reconstruction for a given AMV index $\mathbf{Y}$ performed on a given timeframe $\mathbf{\Gamma}$ using a given statistical regression method M is obtained by applying it with KFCV over the whole learning sample. This reconstruction is associated to a global validation score, calculated as the mean of the individual validation scores obtained for the randomly drawn training and testing splits: $s = \text{avg}(\{s^{(r)}\}_{1 \leq r \leq R})$.

## k-fold cross-validation (KFCV)
Each method requires an optimization of its own set of control parameters $\boldsymbol{\theta}$. To estimate the optimal set of control parameters $\boldsymbol{\theta}_{opt}$ on a given training set $\{\mathbf{x}_{(train)}, \mathbf{Y}_{(train)}\}$, we use a k-fold cross-validation (KFCV) approach.

The KFCV splits the observations into a partition of $K$ groups of the same size (or with approximately the same size if the length of the training set is not divisible by $K$). We denote $\left\{\mathbf{x}_{(k)}, \mathbf{Y}_{(k)}\right\} \forall 1 \leq k \leq K$, which contain only observations for the $k^{th}$ drawn sample. We denote as $\left\{\mathbf{x}_{(-k)}, \mathbf{Y}_{(-k)}\right\}$ the $K$-1 other sets. For all possible values of $\boldsymbol{\theta} \in \Theta$, we scan the $K$ models based on the sets $\{\mathbf{x}_{(-k)}, \mathbf{Y}_{(-k)}\}$. The empirical optimal set of control parameters is obtained by minimizing the averaged Root Mean Squared Errors (RMSE) on the $K$ splits by considering all possible values of $\boldsymbol{\theta}$ (or as much as possible if $\boldsymbol{\theta}$ is defined in a continuous set; see Supplementary Note 1). The optimal KFCV set of control parameters $\boldsymbol{\theta}_{KF}$ is determined by:

$$\hat{\boldsymbol{\theta}}_{opt} = \boldsymbol{\theta}_{KF} = \text{argmin}_{\boldsymbol{\theta} \in \Theta} \frac{1}{K}\sum_{k=1}^{K} \text{RMSE}(\mathbf{Y}_{(k)}, \hat{\mathbf{Y}}_{(k),\boldsymbol{\theta}}) \quad (2)$$

## Nested reconstruction
In this study, the best reconstruction found (defined as the one yielding the best $S_{CE}$ scores) is the reconstruction of the AMV$_F$ index with the random forest method over the period 850–1987 (Supplementary Fig. 6).

Using the same methodological choices (calibration period, AMV definition and reconstruction method), we have then performed a set of 1020 nested reconstructions for the periods 850–1987 to 1869–1987, subsequently using increments of one year for the inferior boundary, which allow us to use an increasing number of proxy records to reconstruct the most recent years. The nested reconstruction (i.e., the reconstruction presented in this study), is obtained by concatenating the first year in each of these 1020 reconstructions.

## Best 30 reconstructions
As additional validation of the results found for the best reconstruction from this study, the 30 best reconstructions out of the 312 have been also produced and studied. Of note, nested reconstructions ranked from the 30th to the 2nd positions have all been produced with a 20-year timestep instead of the one-year one for the best reconstruction in order to largely limit the amount of computing time and energy used. A detailed description of each of these reconstructions and their average $S_{CE}$ score are given in Supplementary Table 4.

## Random forest weights
The weights of the proxy records used for the nested reconstruction are presented in Fig. 2c. Those weights have been calculated using the random forest variable importance. Different importance metrics exist, and for this study, we have selected the commonly used Mean Decrease in Impurity (MDI), also known as Gini importance. The MDI of a given proxy record is calculated as the sum of the number of splits where it is used across the $K$ trees (Supplementary Note 1 for details on the regression methods), proportionally to the numbers of split samples in all trees (cf. Supplementary Note 1). For Fig. 2c, the MDI for each proxy is aggregated over the 1020 reconstructions using a weight of $n/N$, where n is the number of used proxies for a given timestep, and $N = 55$ the total number of proxies used at the end for the reconstruction. Finally, Fig. 2c is computed by calculating the weight of each proxy as a fraction of (i) its calculated MDIs over the timeframes of the nested reconstruction where it is used, and (ii) its total MDI over these timeframes. The same is done in Supplementary Figs. 9 and 10, but each MDI is also averaged over the pseudo-proxy experiments performed on the 12 CESM1-LME members used.

## Validation using random red noise predictors
To verify the relevance of the $S_{CE}$ metric and the quality of our statistical model, a further validation was performed based on randomly generated red noise predictors. The red noise processes are randomly drawn as first-order autoregressive processes, based on the empirical first-order autoregressive coefficient derived from the proxy records. The average reconstruction scores obtained from random red noise predictors are significantly negative ($-0.08$, $P < 0.01$) and significantly lower than the average scores obtained from real proxies over the different periods of the nested reconstruction (0.25, $P < 0.01$, Supplementary Fig. 7).

## Composites of ocean proxy records time series
Since corals are often very short records and ocean sediment cores have too low temporal resolution (preventing them to meet the requirement of being annually resolved), there is a shortage of ocean records contributing to the reconstructions, which are almost exclusively based on terrestrial records. Interestingly, the low-frequency part in the annually resolved reconstructions can be verified against the ocean records from the Ocean2K database that have not been used in the reconstruction. To avoid overfitting, correlation significance shown in Supplementary Fig. 8a is only calculated for the preindustrial period (before 1870) with an AR(1) correction for significance tests to avoid false detections due to the low resolution of some proxies ("Methods"). The composite average time series are performed by multiplying by $-1$ (i.e., calibrated) ocean proxies which have negative correlations with the AMV over the historical period only, such that we mimic a composite-plus-scale reconstruction based on these records only. There is an exception for some ocean records (<10 from Supplementary Fig. 8a) that do not overlap with our reconstruction over the historical period. These time series are therefore multiplied by $-1$ if

their correlations are negative over their overlapping period with the AMV reconstruction.

Detailed descriptions of the results are given in Supplementary Note 2.

## Two-way multimember pseudo-proxy experiments

For the pseudo-proxy experiment, we use 12 members (number 2 to 13) of the National Center of Atmospheric Research (NCAR) Community Earth System Model Lat Millennium Experiment (CESM-LME). Since the calculation of a trend for a given NASST time series is time-dependent, we distinguish the calculation of the model AMV over the preindustrial (PI) period (pre-1870, $AMV_{PI}$) and the historical one ($AMV_H$), notably because anthropogenic forcings were small during the PI period as compared to the recent one (historical). For $AMV_H$, we calculate the AMV by subtracting the ensemble mean of the CMIP5 models, similarly to the real-world reconstruction. The $AMV_F$ is computed by estimating the NASST regression coefficients with the global SST over the PI period.

Each pseudo-proxy mimics a specific real-world proxy by taking, in the model, the variable and season (or annual values) with the largest absolute correlations between the real proxy record and the closest grid points from the CRU TS4 dataset (Supplementary Table 4). Gaps and missing values in real proxy records are also reproduced in the pseudo-proxy time series.

For both pseudo-proxy experiments presented below, and for the sake of reducing computational costs, nested reconstructions have been made with a 20-year timestep for the inferior boundary (from 850–1987 to 1850–1987) instead of the 1-year one that is used for the real reconstructions.

The first pseudo-proxy experiment consists of reconstructing the AMV within model simulations using the same proxy records and RF method as for the real-world reconstruction, and for each timestep of the nested reconstruction. Therefore, the reconstruction scores presented in Supplementary Fig. 9a for each CESM-LME member are averaged over the different timeframes of the nested reconstruction. The correlations are those calculated between the RF-based reconstruction of the model AMV and $AMV_{PI}$. For each member, the proxy record weights are calculated in the same way as for Fig. 2c ("Methods"), and an ensemble average is presented in Supplementary Fig. 9b.

The second pseudo-proxy experiment consists of training RF models directly within the CESM-LME members, in which the pseudo-proxies are selected if the confidence level of the correlation test with the model AMV is above 95%. These trained RF models tailored to the model simulations are then applied to the $AMV_F$ index over the same portion of the historical period than in the real experiment. The obtained reconstructions are then compared to the reconstruction using real-world derived weights in Supplementary Fig. 10c. Since real-world proxy records have been measured with specific units (tree ring MXD, ice core $\delta^{18}O$,...), the pseudo-proxies are rescaled to the mean and the variance of the corresponding real-world proxy. For the same reason, the pseudo-proxy is multiplied by −1 if its correlation with the model AMV has an opposite sign to that of the real-world proxy with the real-world AMV.

For the two pseudo-proxy validations, detailed descriptions of the results are given in Supplementary Note 3.

## Response to volcanic eruptions

To study the potential response of the reconstructed time series (AMV and NASST) we use a superposed epoch analysis. It consists in subtracting for each eruption the years from $N-10$ and $N+20$, where $N$ stands for the year where a given eruption effectively occurred. A composite is then computed as the average of the time series corresponding to the ten strongest eruptions. The 95% confidence level envelope is then calculated for each timestep from the distribution of the surrogate time series.

For the best 30 reconstructions, it is considered that the negative AMV response is significant if at least two years between $N$ and $N + 20$ is lower than the 95% Monte-Carlo envelope.

## Early warning signal test

We base our approach on methods for the detection of incoming climate tipping points using the AR(1) critical slowing down metric. The AMV reconstruction is firstly smoothed using a Kernel Gaussian filtering with a bandwidth of 100 years. The annually resolved AMV is then regressed onto its 100-year filtered version to remove potential long-term trends. The AR(1) coefficients of the residuals from this regression are calculated for different sliding window lengths WL = 200, 250, 300, 350, 400 years (Fig. 6a). The Kendall rank correlation, called Kendall $\tau$, is calculated for each of the AR(1) coefficient series. Contrary to a former study focusing on early warning signal applied to an AMOC collapse in an earth system model of intermediate complexity[38], we cannot use a model-based estimate of the significance of Kendall $\tau$, which is rather calculated using a gaussian approximation as detailed in the "Statistical information" section.

## Statistical information

Figure 1b, c: For each grid point, a two-tailed Student test is applied to the regression coefficients between the corresponding climate variable and the AMV indices. The degrees of freedom are corrected using time series autocorrelations[45].

Figure 6a: The Kendall rank correlation coefficient, or Kendall $\tau$ coefficient, measures the ordinal association between two quantities, here AR(1) coefficients denoted $(\mathbf{x}_i)_{1 \le i \le n}$ here, and time denoted $(\mathbf{y}_i)_{1 \le i \le n}$. The statistic is given by:

$$\tau = \frac{n_c - n_d}{n_0} \tag{3}$$

Where, considering $(\mathbf{x}_1, \mathbf{y}_1), (\mathbf{x}_2, \mathbf{y}_2), ..., (\mathbf{x}_n, \mathbf{y}_n)$ the ensemble of joint pairs:

$$n_c = \mathrm{card}_{i \ne j}\left\{ \left\{ \mathbf{x}_i > \mathbf{x}_j \cap \mathbf{y}_i > \mathbf{y}_j \right\} \cup \left\{ \mathbf{x}_i < \mathbf{x}_j \cap \mathbf{y}_i < \mathbf{y}_j \right\} \right\}, (i,j) \in [[1,n]]^2 \tag{4}$$

$$n_d = \mathrm{card}_{i \ne j}\left\{ \left\{ \mathbf{x}_i > \mathbf{x}_j \cap \mathbf{y}_i < \mathbf{y}_j \right\} \cup \left\{ \mathbf{x}_i < \mathbf{x}_j \cap \mathbf{y}_i > \mathbf{y}_j \right\} \right\}, (i,j) \in [[1,n]]^2 \tag{5}$$

$$n_0 = \frac{n(n-1)}{2} \tag{6}$$

For large sample ($n > 50$), as in this study, the distribution is approximated with a Gaussian distribution of mean 0 and variance $\frac{2(2n+5)}{9n(n-1)}$, under the null hypothesis $H_0$: "$\tau = 0$" which is tested against the alternative hypothesis $H_1$: "$\tau \ne 0$". The $P$ value (shown in Fig. 6a) of the test is deduced from the quantiles of this distribution.

Unless stated otherwise correlation are tested using a Student $t$ test for correlation, with corrected degrees of freedom using time series autocorrelation, as shown for instance in Supplementary Fig. 8a.

For all boxplots of the study, the median is shown as a heavy darkline. Boxplots edges give first and third quartiles. Boxplot "whiskers" gives the full range without including outliers, which are not shown here for better graphical representations. A point from a boxplot is here considered as an outlier when it is outside 1.5 times the interquartile range above the upper quartile and below the lower quartile.

## Data availability

The whole data generated and treated in this study have been deposited on a Zenodo repository: https://zenodo.org/record/4896670#.YLjdOS2w3dc. The source and treated data to compute figures in this study are provided in the Source Data file. The generated data in this study are provided in the Data file. Simulation outputs for the LME experiments can be accessed from the NCAR Climate Data gateway website: https://www.earthsystemgrid.org/. The CRU TS4 datasets can be accessed from the CRU data download webpage: https://crudata.uea.ac.uk/cru/data/hrg/#current. The HadISST data can be accessed from the Met Office Hadley Center website: https://www.metoffice.gov.uk/hadobs/hadisst/. For the historical experiments of CMIP5 and CMIP6 model outputs, data can be accessed from the ESGF website: https://esgf-node.llnl.gov/projects/esgf-llnl/. Source data are provided with this paper.

## Code availability

All codes needed to reproduce this study are publicly available on the following Zenodo link: https://zenodo.org/record/4896670#.YLjdOS2w3dc. Codes have been made using bash UNIX and R languages. R packages are listed in Supplementary Table 8.

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

## Acknowledgements

This study benefited from the IPSL Prodiguer-Ciclad and Camelot supercomputing facilities, supported by CNRS, UPMC Labex L-IPSL. This work was also sponsored by NWO Exact and Natural Sciences for the use of SurfSARA supercomputing facilities (Amsterdam) under project 2020.022. Authors were also funded by EU-H2020 TiPES (Grant no. 820970, S.M., contribution no. 164), Blue Action (Grant Agreement no. 727852, D.S., G.G, G.M., and J.M.), EUCP (Grant Agreement no 776613, D.S. and J.M.), ROADMAP (Grant Agreement no. 116020, J.M., G.M., and G.G.) ARCHANGE (ANR-18-MPGA-0001, J.M. and G.G.) research programmes.

## Author contributions

S.M. has performed the different reconstructions, statistical analysis, figures, and the pseudo-proxy experiment of this study. S.M. has mainly written the manuscript with the important participation of D.S., J.M., P.O., and G.G. D.S., J.M., P.O., G.G., M.K., and G.M. have contributed to the results assessments, the manuscript writing, and have made suggestions to set up the manuscript's guiding thread.

## Competing interests

The authors declare no competing interests.
