## [Peer Review File · Nature Communications]

Early warning signal for a tipping point suggested by a millennial Atlantic Multidecadal Variability reconstructionReviewers' Comments:

Reviewer #1:

Remarks to the Author:

The authors have made significant improvements to their manuscript since its original submission, and I am satisfied in the way they have dealt with my previous concerns and suggestions - either by corrections to the manuscript or, in a few cases, through thorough explanations and arguments in their rebuttal letter.

Thus, I only have a few minor technical suggestions (listed below) and believe that the manuscript is ready for publication.

Line 60-61: Add the reference (22) after "last millennium" to make it fully clear what "This study" refers to.

Figure: The maps are very small. I understand that the authors have group the maps according to season and parameter, but I strongly suggest that they still try to work on finding a way to increase the size of the maps. Also, the text of the maps is so small that it is nearly unreadable.

Figures 4+5: The "title" (bold part of the captions) of the figures are the same. The differences between the two figures should be explained in the titles by adding a short explanation highlighting the specifics of the figure. For Fig 5 it could e.g. be "Model to proxy PPE validation for AMVF".

I would still have preferred the authors to have added "also called the Atlantic Multidecadal Oscillation" in line 36-37 after the AMV is mentioned the first time, but the authors have made a choice here and argue for this in their rebuttal letter, and since they refer to work using the term AMO, I accept their choice.

Note that I am not been able to evaluate the technical parts of the model work.

/Marit-Solveig Seidenkrantz

Reviewer #2:

Remarks to the Author:

The manuscript "A 1150-year-long AMV reconstruction suggests early warning for a North Atlantic climate tipping point" presents a comparison of various methods to reconstruct the Atlantic Multidecadal Variability from 850 C.E. to the present. The authors use various proxy records from around the globe to establish a longer record for the AMV than the historical one. This is particularly important for establishing long-term oscillatory behaviour. Using proxy records the authors deem relevant through temporal resolution, location, and correlation with the historical record, they produce 312 reconstructions based on various AMV definitions, temporal windows, and regression methods. The reconstructions are evaluated using the Coefficient of Efficiency (CE) metric and the one with the overall max CE metric is chosen for analysis. The chosen reconstruction displays multidecadal variability within the band of 20-90 years and also signals the approach to a tipping point.

Overall I feel some improvements to the presentation of the results are necessary to clarify the message of the manuscript. The motivation for each step of the analysis is lost in the text and does not form a coherent story that is easy to follow. The main conclusions regarding the AMV are rushed and also not made general enough given the analysis set-up. I offer some general comments below, as well as more specific points and technical corrections.

General comments:

1. In general I find the manuscript confusing to read. The Results sections reads more like a Methods section, while corresponding parts of the Methods section are quite clear and concise. I would suggest the authors try to rewrite the both of these, putting unnecessary details of the all the calculations in Methods and only discussing the general set-up, outcomes, and interpretations of the outcomes in the main body of text.

2. It is not clear why there are multiple AMV indices calculated. The authors nicely explain the difference between the way in which they are produced, however there is no discussion of the resulting indices. It would be nice if a comparison of the indices was added through more discussion of Figure 1. This would help to motivate using each of them in the reconstructions (i.e. if they are very similar than one would not need to consider all of them for the reconstructions). Please specify the major differences in the resulting indices and what one intends to gain by considering all three.

3. The conditions on the proxy records needs to be explained more and defended, particularly when it comes to exclusion based on location. In the results section there are proxy records included which are far from the North Atlantic and teleconnections are cited in the defence for inclusion. However, in the methods section all proxy records in the Southern Hemisphere are excluded from the start. Is there a specific reason teleconnections to the Southern Hemisphere are not being considered? As the AMV is an ocean mode and not an atmospheric one, there is even more reason to believe it can impact conditions in the Southern Hemisphere through ocean circulation patterns. I suggest the authors include such records initially and then accept or reject based on the other two conditions (temporal resolution and correlation), or otherwise provide an more in-depth discussion as to why proxies in the Southern Hemisphere can be neglected.

4. It is common practice that sentences do not begin with an acronym, parameter, or variable. I suggest that the authors take note of this throughout the manuscript and change where necessary (take particular note in the Methods sections).

5. The abstract seems to emphasize more the periodicity of AMV oscillations and a potential tipping point, but I found that the text was mainly centred around the reconstruction. I think the authors should clarify the focus of the paper. Additionally, it is not clear how choosing a different reconstruction would affect the wavelet and early warning signal analyses. The authors choose the reconstruction with the highest max CE value, but to my knowledge it is not stated how far away this value is from other values. Looking at the box and whisker plots it does not seem to be very far. I think it would improve the paper if the authors could comment on the results when using other reconstructions with high CE values. One would hope that small changes in the hyperparameters (length of window, AMV definition, regression method) would not impact results significantly. While the results of analyses on other reconstructions do not necessarily need to be shown they could at least be discussed.

Specific comments:

Lines 147-149: It is stated that "moving windows of one year" are used for the temporal reconstruction. This to me implies that the windows are all uniform length of one year. This is not what is described in the methods section which states that the windows are all of varying lengths, overlapping times, and have different upper bounds in intervals of one year. Please adjust the language here to clarify.

Line 155: Similar to previous comment, the statement "only differing in the last year covered by the proxies" is misleading as it implies all the windows are of the same length with a different last year. Perhaps change to "only differing in the upper bound on the time window"

Lines 170-171: Can you explain what is meant by "the highest averaged score of all reconstructions"?

In fact, the connection between individual reconstructions and averaged reconstructions needs to be elaborated on a bit more in this section.

Lines 194-196: Please expand on the discussion of the teleconnection with the Eastern Pakistan/Tibetan Plateau and elaborate on the effect of the AMV related to that specific region (extending from just mentioning large scale pressure gradients).

Line 287: The term "ongoing bifurcation" is not defined and I am not quite sure what it is describing. Is it implied that the bifurcation is a continuous process? I am not sure I agree with this interpretation as I am unaware of any theory regarding ongoing bifurcations. In this context perhaps "approaching bifurcation" is better terminology.

Lines 297-298: "This suggests that more than 40%..." - unclear where the 40% came from.

Lines 301-305: "A plausible explanation..." - I do not fully understand the theory suggested here. Perhaps the sentence just needs to be reworded to provide clarity. It seems as though you are defending why the NASST reconstruction score is less skillful than that of the AMV, whereas the previous sentence states the opposite.

Lines 684-685: There is already a section on NASST reconstruction. This sentences seems out of place and unnecessary.

Line 718: Please explain what is meant by "random splits" here.

Lines 722-723: "Cross validation methods..." needs some references.

Line 732: It is not explained how \hat{Y} is computed.

Lines 742-743: "The nested reconstruction ..." - This statement seems to contradict the previous sentence. Are you using only the first year of each reconstruction or are you using the same year in overlapping reconstructions? The explanation of the nested reconstruction in general was quite confusing for me.

Lines 752-754: " n/N where n is the number..." - Would this not be a number greater than one rather than a percentage? Either that or I am misunderstanding what is meant by available proxies and total number used. You give a sample N value but no sample n value. Perhaps adding two sample values would be more informative.

Lines 754-757: Interchanging between "MDI" and "importance" is confusing here. Can you retain the same terminology if you are referring to the same measure? I am also unsure of how the percentage is actually calculated. What does "previously calculated importance" mean within this process?

Line 761: "no method to remove the forced variability is applied" - Why have you made this choice? This is different from the choice you made for the AMV so must be motivated.

Line 762-763: "The final NASST reconstruction is obtained..." - How? Using max CE value? Is there a figure that shows this? Please elaborate.

Line 788: "the same is applied" - The same what is applied?

Lines 860-864: Does this need to be a separate section as the Fig 1b,c section? Can you combine the two?

Fig 1 caption: I do not understand what is meant by "from a) and". Additionally, what does "c-e"

mean? Only those two plots? c,d, and e? I would think this refers to all plots but a.

Fig 2b: I do not see the lake sediment, ice core, or coral in this figure.

Fig 3 caption: Please explain the orange and green lines of 3b in the caption.

Fig 4 caption: Should mention somewhere here that 4b is an ensemble average.

Extended Fig. 2: The subplots are not labeled a-f. Also, perhaps mention the difference in the columns first and then the difference by row, with the number of samples in each denoted where appropriate.

Technical suggestions:

Line 20: "controversial" to "highly debated"

Lines 36-37: add "also known as the Atlantic Multidecadal Oscillation" after introduction of AMV

Line 99: "consistently" to "consistent"

Line 233: "methods" should be capital (and throughout the manuscript)

Line 294: "than" to "as"

Line 309 "than" to "as"

Line 324: "a" to "are"

Line 552: "kipping" to "tipping"

Line 690: Remove "as" after "denote"

Line 692: The k in the first inequality should be a j

Line 701: Indent here

Line 703: "involves" to "requires"

Line 706: The overbar is more commonly used here rather than the underscore when notating a sum.

Line 764: "an increment of one year" to "yearly increments"

Line 790: Indent here

Line 796: Indent here

Line 860: "than" to "as"

Reviewer #3:

Remarks to the Author:

This study uses a disparate set of proxies to estimate the past behavior of the Atlantic Multidecadal Variability. While an AMV reconstruction would be most welcome, as the debate continues to rage about its origin and existence and role in past and future climate variability and change, I am

concerned about how proxies have been selected for the reconstruction and how realistic the inclusion of these (and exclusion of others) is. I was not a reviewer on the original manuscript, but find myself in agreement with several of the concerns raised in the previous review. To summarize:

1. Why use several different AMV indices when they are functionally the same (e.g. Figure 1a)? Why additionally smooth the series and reduce the degrees of freedom so much, particularly given the other challenges with short instrumental records and proxies records that end in the 1980s and 1990s? It would also be useful to compare the AMV indices used here to the AMO itself, since the AMV indices that are a target of this paper looks a lot like the AMO still - eg. negative from 1900 to 1930, positive from 1930 to 1960, negative again from 1960 to 2000, and positive afterwards. Indeed, it isn't clear what has been accomplished by in theory removing the forced signal.

2. The study starts with a very large number of proxy data, which is reduced to approximately ~15% of the initial total. Worryingly, there seems to be little that explains the ultimate importance of these proxies - the majority are from Asia, not Europe or eastern North America, as one might expect. Adjacent or co-located proxy records sometimes enter the reconstruction model and sometimes don't. For instance, MOR2JU is part of the reconstruction, but not the co-located MOR3JU and MOR4JU? (Disturbingly, the authors reveal that a single record from Pakistan MOR1JU weighted heavily in the reconstruction, so they removed it!) Why would the Maiana coral record from the western Pacific enter into a reconstruction of North Atlantic SSTs? In California, why would the Fish Creek Trail record from Jeffrey Pine enter the model but no other chronologies from California or the western United States? The final list of proxies looks like a random collection but with a weight toward interior Asia, with some exceptions, raising concerns about whether using such a large predictor dataset results in a random selection of proxies. Even in Asia, the sign of the relationship varies between proxies. Indeed, the heavy RF weights associated with sites quite distant from the Atlantic itself raise questions about the proxy inclusion - why wouldn't sites under the more direct influence of the Atlantic show the most importance? The correlations between AMV and proxies reported in Extended Data Table 3 are also quite low (e.g.), especially considering the smoothness of the predictand AMV series. Were the p-values adjusted for reduced degrees of freedom? The authors spend a good part of the manuscript (L186 to L219) in post hoc justification of these unexpected proxy selections.

2. The authors say the following: 'An important preprocessing step before selecting the proxies is to remove in each proxy record from P2k+ (see Methods) the forced variability of NASST, using linear regression starting in 1870' but nowhere in the methods can I find a description of how this was done. Are the authors suggesting they removed the annual forced NASST signal from local proxies? This seems problematic, as one would expect the local proxy signal to include the direct forcing (especially for temperature) independent of what the Atlantic is doing.

3. Were the CE scores here benchmarked against red noise? As Wahl and Smerdon 2012 suggest, when degrees of freedom are low and there is the potential for strong red noise, a null test of the CE scores may be warranted. CE is low although in places significant here, but given that a smoothed AMV has relatively few degrees of freedom and the validation period is quite short, an evaluation of whether $CE > 0$ is really the cutoff for significance in this metric would be useful.

4. It isn't clear why the study selects marine proxies for an ad hoc validation - confusingly, a validation of AMV uses corals from the central Pacific and marine records from the southern hemisphere - it is difficult to see, particularly given the level of smoothing, how this proves and secures validation.

General authors comments

We thank the reviewers for their constructive and useful comments that clearly helped to improve the quality of our manuscript. We believe that the reviewers' concerns have been fully addressed and we hope reviewers will be convinced by the new manuscript of our study. In the following document, questions from the reviewers are written as normal light text, while the **authors responses are written in bold**

Reviewer #1 (Remarks to the Author):

The authors have made significant improvements to their manuscript since its original submission, and I am satisfied in the way they have dealt with my previous concerns and suggestions - either by corrections to the manuscript or, in a few cases, through thorough explanations and arguments in their rebuttal letter.

We thank the reviewer for her overall good appreciation of our manuscript, and for her useful comments and suggestions in her review reports.

Thus, I only have a few minor technical suggestions (listed below) and believe that the manuscript is ready for publication.

Line 60-61: Add the reference (22) after "last millennium" to make it fully clear what "This study" refers to.

The reference has been added to this sentence.

Figure: The maps are very small. I understand that the authors have group the maps according to season and parameter, but I strongly suggest that they still try to work on finding a way to increase the size of the maps. Also, the text of the maps is so small that it is nearly unreadable.

This figure was indeed hard to read in its former form. We have improved the quality of the figure such that all the information is more visible.

Figures 4+5: The "title" (bold part of the captions) of the figures are the same. The differences between the two figures should be explained in the titles by adding a short explanation highlighting the specifics of the figure. For Fig 5 it could e.g. be "Model to proxy PPE validation for AMVF".

We have modified and clarified the titles of both figures 4 and 5 following the reviewer's instruction.

I would still have preferred the authors to have added "also called the Atlantic Multidecadal Oscillation" in line 36-37 after the AMV is mentioned the first time, but the authors have made a choice here and argue for this in their rebuttal letter, and since they refer to work using the term AMO, I accept their choice.

We now mention the fact that the mode called AMV in our study has long been called the AMO.

Note that I am not been able to evaluate the technical parts of the model work.

/Marit-Solveig Seidenkrantz

Reviewer #2 (Remarks to the Author):

The manuscript "A 1150-year-long AMV reconstruction suggests early warning for a North Atlantic climate tipping point" presents a comparison of various methods to reconstruct the Atlantic Multidecadal Variability from 850 C.E. to the present. The authors use various proxy records from around the globe to establish a longer record for the AMV than the historical one. This is particularly important for establishing long-term oscillatory behaviour. Using proxy records the authors deem relevant through temporal resolution, location, and correlation with the historical record, they produce 312 reconstructions based on various AMV definitions, temporal windows, and regression methods. The reconstructions are evaluated using the Coefficient of Efficiency (CE) metric and the one with the overall max CE metric is chosen for analysis. The chosen reconstruction displays multidecadal variability within the band of 20-90 years and also signals the approach to a tipping point.

Overall I feel some improvements to the presentation of the results are necessary to clarify the message of the manuscript. The motivation for each step of the analysis is lost in the text and does not form a coherent story that is easy to follow. The main conclusions regarding the AMV are rushed and also not made general enough given the analysis set-up. I offer some general comments below, as well as more specific points and technical corrections.

We thank the reviewer for the new comments that definitely helped us to significantly improve the new version of the manuscript and hopefully clarify the conclusions.

General comments:

1. In general I find the manuscript confusing to read. The Results sections reads more like a Methods section, while corresponding parts of the Methods section are quite clear and concise. I would suggest the authors try to rewrite the both of these, putting unnecessary details of the all the calculations in Methods and only discussing the general set-up, outcomes, and interpretations of the outcomes in the main body of text.

We have carefully re-worked the narrative and structure of the paper in order to make it clearer to a wide audience. We have tried to further limit the description of technical details in the main text, and to only keep those aspects that are essential to understand the novelty and benefits of our reconstruction methods, as well as the importance of the independent ways to validate the results.

2. It is not clear why there are multiple AMV indices calculated. The authors nicely explain the difference between the way in which they are produced, however there is no discussion of the resulting indices. It would be nice if a comparison of the indices was added through more discussion of Figure 1. This would help to motivate using each of

them in the reconstructions (i.e. if they are very similar than one would not need to consider all of them for the reconstructions). Please specify the major differences in the resulting indices and what one intends to gain by considering all three.

We agree with the reviewer that it might appear strange to use three AMV indices while they all are dedicated to describing variations of the same phenomenon. Although very close, the paired correlations indicate substantial differences between them (Fig. 1), where $\text{cor}(\text{AMV}_{\text{TS}}, \text{AMV}_{\text{T}})=0.63$, $\text{cor}(\text{AMV}_{\text{F}}, \text{AMV}_{\text{T}})=0.90$, $\text{cor}(\text{AMV}_{\text{TS}}, \text{AMV}_{\text{F}})=0.76$. In fact the question of how to remove the external forcing signal in the variations of North Atlantic SST is difficult, notably because of the complex nature in the regional aerosol forcing. Thus, several methods have been proposed which led to the three indices used. Each method has its pros and cons, and while providing relatively close indexes, they exhibit slight differences as highlighted above. For instance, AMV_{T} uses an estimation of the external forcings' influence on NASST using an ensemble of model simulations while the two others only need observations to be calculated. To have robust reconstructions which include this uncertainty, we thus decided to use the three main indexes proposed in the existing literature. However, we agree with the reviewer that whatever the index used, the obtained results should not be very different if our reconstruction method is robust. Or, if there are differences, they should be highlighted in the manuscript. For these reasons, we keep presenting results and reconstruction validation for the reconstruction with the highest CE scores in the new version of the manuscript. However, to address the reviewer comment, we also test the sensitivity of the results presented (response to forcings, spectrum and early warning signal) to the reconstruction procedure. For doing so, we have added supplementary figures and tables in which the study results are also tested for the 30 best reconstructions out of the initial 312 (including at least one reconstruction for each of the studied indices) in addition to those for the best one presented in the main text. This sensitivity analysis confirms that the main results of our study are consistent regardless of the index or the reconstruction method considered.

3. The conditions on the proxy records needs to be explained more and defended, particularly when it comes to exclusion based on location. In the results section there are proxy records included which are far from the North Atlantic and teleconnections are cited in the defence for inclusion. However, in the methods section all proxy records in the Southern Hemisphere are excluded from the start. Is there a specific reason teleconnections to the Southern Hemisphere are not being considered? As the AMV is an ocean mode and not an atmospheric one, there is even more reason to believe it can impact conditions in the Southern Hemisphere through ocean circulation patterns. I suggest the authors include such records initially and then accept or reject based on the other two conditions (temporal resolution and correlation), or otherwise provide an more in-depth discussion as to why proxies in the Southern Hemisphere can be neglected.

Southern Hemisphere proxy records were included in the reconstructions of the first version of this manuscript, and were later excluded to address the concerns of one of the reviewers in the first round of reviews. We have however tested the sensitivity of the reconstruction to the inclusion of the Southern Hemisphere proxies and found that those proxy records had relatively small weights in the reconstructions which yielded very similar results in terms of spectrum, response to external forcings and early warning signals. This

was in turn expected since in the first version of this manuscript, where southern hemisphere records were included, they did not have large weights and represented a small sample of the whole proxy records dataset used. Of note, numerous papers argue that AMV mostly impacts northern hemisphere regions (Knight et al. 2005, Zhang et al. 2019), which at least means that the inclusion of southern hemisphere proxy records is expected to barely be able to improve the final reconstruction and to significantly modify the latter.

4. It is common practice that sentences do not begin with an acronym, parameter, or variable. I suggest that the authors take note of this throughout the manuscript and change where necessary (take particular note in the Methods sections).

All the sentences starting with an acronym or a variable have been rewritten in the new version of the manuscript and we thank the reviewer for pointing out this problem.

5. The abstract seems to emphasize more the periodicity of AMV oscillations and a potential tipping point, but I found that the text was mainly centred around the reconstruction. I think the authors should clarify the focus of the paper. Additionally, it is not clear how choosing a different reconstruction would affect the wavelet and early warning signal analyses. The authors choose the reconstruction with the highest max CE value, but to my knowledge it is not stated how far away this value is from other values. Looking at the box and whisker plots it does not seem to be very far. I think it would improve the paper if the authors could comment on the results when using other reconstructions with high CE values. One would hope that small changes in the hyperparameters (length of window, AMV definition, regression method) would not impact results significantly. While the results of analyses on other reconstructions do not necessarily need to be shown they could at least be discussed.

We agree with the reviewer that the main manuscript was possibly not well balanced and was not reflecting sufficiently the main results highlighted in the abstract. As a consequence we have re-written the main manuscript, expanding the discussion of the scientific results given by the reconstruction itself. In addition, to illustrate the sensitivity of the results to the methodological choices, we have included some new analyses with the 30 best performing reconstructions, as already mentioned in our response to comment #2.

Specific comments:

Lines 147-149: It is stated that "moving windows of one year" are used for the temporal reconstruction. This to me implies that the windows are all uniform length of one year. This is not what is described in the methods section which states that the windows are all of varying lengths, overlapping times, and have different upper bounds in intervals of one year. Please adjust the language here to clarify.

The latter sentence has been rewritten and clarified to fit with the Methods description.

Line 155: Similar to previous comment, the statement "only differing in the last year covered by the proxies" is misleading as it implies all the windows are of the same

length with a different last year. Perhaps change to "only differing in the upper bound on the time window"

The modification has been made.

Lines 170-171: Can you explain what is meant by "the highest averaged score of all reconstructions"? In fact, the connection between individual reconstructions and averaged reconstructions needs to be elaborated on a bit more in this section.

We have here clarified that the highest averaged CE scores are obtained over the 30 training/testing sample splits essentially made for this purpose.

Lines 194-196: Please expand on the discussion of the teleconnection with the Eastern Pakistan/Tibetan Plateau and elaborate on the effect of the AMV related to that specific region (extending from just mentioning large scale pressure gradients).

More details and references have been added concerning AMV teleconnections over the different Asian regions where proxy records are picked.

Line 287: The term "ongoing bifurcation" is not defined and I am not quite sure what it is describing. Is it implied that the bifurcation is a continuous process? I am not sure I agree with this interpretation as I am unaware of any theory regarding ongoing bifurcations. In this context perhaps "approaching bifurcation" is better terminology.

We agree with the reviewer that this statement is not clear. We have changed the terminology in the new version of the manuscript following the reviewer's instructions. More generally, the different parts concerning the tipping point test and the implications of the results are discussed with more details and clarity in the new version of the manuscript.

Lines 297-298: "This suggests that more than 40%..." - unclear where the 40% came from.

This sentence have been removed in the new version of the manuscript

Lines 301-305: "A plausible explanation..." - I do not fully understand the theory suggested here. Perhaps the sentence just needs to be reworded to provide clarity. It seems as though you are defending why the NASST reconstruction score is less skillful than that of the AMV, whereas the previous sentence states the opposite.

This statement has been clarified and reworded in the new version of the manuscript.

Lines 684-685: There is already a section on NASST reconstruction. This sentences seems out of place and unnecessary.

This sentence have been removed in the new version of the manuscript.

Line 718: Please explain what is meant by "random splits" here.

This part of the text has been clarified.

Lines 722-723: "Cross validation methods..." needs some references.

A reference has been added (reference number 44).

Line 732: It is not explained how \hat{Y} is computed.

This part of the text has been clarified.

Lines 742-743: "The nested reconstruction ..." - This statement seems to contradict the previous sentence. Are you using only the first year of each reconstruction or are you using the same year in overlapping reconstructions? The explanation of the nested reconstruction in general was quite confusing for me.

The definition of nested reconstruction and the way it is computed is now explained more clearly in the Methods section.

Lines 752-754: " n/N where n is the number..." - Would this not be a number greater than one rather than a percentage? Either that or I am misunderstanding what is meant by available proxies and total number used. You give a sample N value but no sample n value. Perhaps adding two sample values would be more informative.

Here, $n \leq N$ with n increasing with time (since more proxy records are used). This scaling of weights is used to avoid accounting too much weights to longer proxy records (that effectively have large weights for longest (oldest) timeframes of the reconstruction since there are a few of them).

Lines 754-757: Interchanging between "MDI" and "importance" is confusing here. Can you retain the same terminology if you are referring to the same measure? I am also unsure of how the percentage is actually calculated. What does "previously calculated importance" mean within this process?

Here, MDI is one of the existing metrics to measure the "importance" of variables of the RF method. It has been clarified in the Methods section. The "previously calculated" meant the MDI statistics (relative to each proxy) calculated for the timeframes of the nested reconstruction where a particular proxy record is used. The purpose is to normalize the mean weight of each proxy by its length (since they are not used the same number of times within the nested reconstruction process) such that this weight does not depend on the length of the proxy. We have re-written this Methods section to try to clarify this aspect.

Line 761: "no method to remove the forced variability is applied" - Why have you made this choice? This is different from the choice you made for the AMV so must be motivated.

AMV indices are constructed with an attempted removal of the external forcings. On the other hand, NASST, as the spatially-averaged SST anomalies in the North Atlantic, thus mixes variations responding to external forcings and others to internal climate processes. This is the reason why when reconstructing the AMV, where the external forcing is removed, we also try to remove it from proxies. While for the NASST, we use the raw timeseries of proxies instead, such that no externally forced variations are removed from predictors (raw proxies) and the predictand (NASST) in that case. This is explained in clearer terms in the new version of the manuscript.

Line 762-763: "The final NASST reconstruction is obtained..." - How? Using max CE value? Is there a Figure that shows this? Please elaborate.

Indeed, the reviewer is right to say that some information is missing here. An additional supplementary figure, giving the scores obtained for the different NASST reconstructions compared similarly to Supplementary Fig. 2 (see Fig. R1 below), is now accompanying the new version of the manuscript.

Fig. R1. Score by level of inputs for the 104 NASST reconstructions. Coefficient of Efficiency (CE) scores for the 104 reconstructions compared in this study for the different sources of methodological choices (regression method and reconstruction frame). **a** and **b** give the CE scores by regression methods (104/4=26 reconstructions by method). **c** and **d** give the CE scores by superior boundary of the reconstruction window (104/26=12 final reconstructions by window). **a** and **c** give the CE scores for all the training splits (104*30=3,120 scores, respectively 780 and 120 scores by boxplot for the three panels) **b** and **d** give CE scores for each final reconstruction as the averages of the 30 corresponding individual CE scores (104 average scores). Red dots indicate the highest average score obtained for the particular level of input. For all boxplots, medians are shown as heavy dark lines. Boxplots edges give first and third quartiles. Boxplot "whiskers" give the 10%-90% range. Outliers are not shown. A point from a boxplot is here considered as an outlier when it is outside 1.5 times the interquartile range above the upper quartile and below the lower quartile.

Line 788: "the same is applied" - The same what is applied?

We have clarified here that the same methodology of model selection (by max CE) is applied.

Lines 860-864: Does this need to be a separate section as the Fig 1b,c section? Can you combine the two?

This part no longer exists in the new version of the manuscript.

Fig 1 caption: I do not understand what is meant by "from a) and". Additionally, what does "c-e-" mean? Only those two plots? c,d, and e? I would think this refers to all plots but a.

The "c-e-" definitely was a typo and it has been corrected. In the new version, we have turned all mentions to panels in figures captions from "from a) and" to "from panel a) and" in order to clarify the figure descriptions.

Fig 2b: I do not see the lake sediment, ice core, or coral in this figure.

The amount of coral and sediment lake proxies was indeed too small to be well seen in the figure. But there are indeed no ice cores used, but a document proxy instead. This figure has been modified such that we can see these types of proxy records as clearly as possible, although it is still a bit hard to distinguish because their amount is extremely small as compared to the amount of tree rings used.

Fig 3 caption: Please explain the orange and green lines of 3b in the caption.

This is now clearly explained in the figure caption.

Fig 4 caption: Should mention somewhere here that 4b is an ensemble average.

This information has been added in the figure caption.

Extended Fig. 2: The subplots are not labeled a-f. Also, perhaps mention the difference in the columns first and then the difference by row, with the number of samples in each denoted where appropriate.

The missing labels for the panels have been added. And the column and rows of the figures are now described with more details.

Technical suggestions:

The below reviewer's comments/suggestions on the English writing have all been addressed and we thank her/him for it.

Line 20: "controversial" to "highly debated"

Lines 36-37: add "also known as the Atlantic Multidecadal Oscillation" after introduction of AMV

Line 99: "consistently" to "consistent"

Line 233: "methods" should be capital (and throughout the manuscript)

Line 294: "than" to "as"

Line 309 "than" to "as"

Line 324: "a" to "are"

Line 552: "kipping" to "tipping"

Line 690: Remove "as" after "denote"

Line 692: The k in the first inequality should be a j

Line 701: Indent here

Line 703: "involves" to "requires"

Line 706: The overbar is more commonly used here rather than the underscore when notating a sum.

Line 764: "an increment of one year" to "yearly increments"

Line 790: Indent here

Line 796: Indent here

Line 860: "than" to "as"

Reviewer #3 (Remarks to the Author):

This study uses a disparate set of proxies to estimate the past behavior of the Atlantic Multidecadal Variability. While an AMV reconstruction would be most welcome, as the debate continues to rage about its origin and existence and role in past and future climate variability and change, I am concerned about how proxies have been selected for the reconstruction and how realistic the inclusion of these (and exclusion of others) is. I was not a reviewer on the original manuscript, but find myself in agreement with several of the concerns raised in the previous review.

We thank the reviewer for her/his useful and relevant comments on our manuscript. In general, we have performed new analysis and brought new references to support our results. We hope that the reviewer will appreciate the changes we have made to address her/his suggestions

To summarize:

1. Why use several different AMV indices when they are functionally the same (e.g. Figure 1a)?

As we explained to reviewer #2, we use three AMV indices because the question of how to remove the external forcing signal in the variations of North Atlantic SST is non trivial, notably because of the complex and regional aerosol forcing, which can introduce differences between the indices which would thus affect the reconstructions. Several methods have been proposed which led to the three indexes used. Each method has its pros and cons, and while most of the variability is common to the three indices, they can also exhibit non-negligible differences. Indeed, the paired correlations indicate the largest differences in particular with the AMV_{TS} index (Fig. 1): $cor(AMV_{TS}, AMV_T)=0.63$, $cor(AMV_F, AMV_T)=0.90$, $cor(AMV_{TS}, AMV_F)=0.76$. To have robust reconstructions which include this uncertainty, we thus decided to use the three indexes proposed, without making *a priori* assumptions of which one is better. However, we agree with the reviewer that whatever the index used, the obtained results should not be very different if our reconstruction method is robust. For these reasons, we keep presenting results and reconstruction validation for the reconstruction with the highest CE scores in the new version of the manuscript. However, to address the reviewer comment, we also test the sensitivity of the results presented (response to forcings, spectrum and early warning signale) to the reconstruction setup. For doing so, we have added supplementary figures and tables in which the study results are also tested for the 30 best reconstructions out of the initial 312 (including at least one reconstruction for each of the studied indices) in addition to those for the best one presented in the main text. We believe that the consistency of our study results regardless of the index or the reconstruction method used will be reinforced with this additional sensitivity analysis.

Why additionally smooth the series and reduce the degrees of freedom so much, particularly given the other challenges with short instrumental records and proxies records that end in the 1980s and 1990s?

We did not smooth the AMV series for the calibration of statistical models for the reconstruction such that the degrees of freedom is not too much reduced. This was apparently not stated sufficiently clearly and has been rewritten in the main text. However, because AMV/AMO is usually considered as a multi-decadal mode of variability and smoothed at 10 years to better highlight this time scale, we showed the reconstructions with this smoothing although they were produced with annually-resolved proxies and AMV indices (with removed external forcings).

For the shortness of records, it is worth noting that the records used for statistical climate reconstructions need to be gap-free. Then, if we use more years of observations, there will mechanically be less proxy records available for the most recent years. However, there is no way to *a priori* know what will be the right balance between the number of observation years and the number of proxy records. This is why there are 26 timeframes over which we try constructing the reconstruction models: from 1870-1975 to 1870-2000 (on top of 3 indices and 4 regression methods). It is

clear on Supplementary Fig. 2 (see Fig. R2 below) that, on average, increasing the number of years used for the statistical modelling gives much lower scores after the 1990s than for the late 1980s (because much less gap-free records are available for most recent years).

Fig. R2 (Supplementary Fig. 2). **Score by level of inputs for the 312 reconstructions.** Coefficient of Efficiency (CE) scores for the 312 reconstructions compared in this study for the different sources of methodological choices (regression method, reconstruction frame and AMV index). **a** and **b** give the CE scores by regression methods (312/4=78 reconstructions by method). **c** and **d** give the CE scores by superior boundary of the reconstruction window (312/26=12 final reconstructions by window). **e** and **f** give the CE scores by AMV index (312/3=104 final reconstructions by index). **a**, **c** and **e** give the CE scores for all the training splits (312*30=9,360 scores, respectively 2340, 360, and 3120 scores by boxplot for the three panels). **b**, **d** and **f** give CE scores for each final reconstruction as the averages of the 30 corresponding individual CE scores (312 average scores). Red dots indicate the highest average score obtained for the particular level of input. For all boxplots, medians are shown as heavy dark lines. Boxplots edges give first and third quartiles. Boxplot "whiskers" give the 10%-90% range. Outliers are not shown. A point from a boxplot is here considered as an outlier when it is outside 1.5 times the interquartile range above the upper quartile and below the lower quartile.

It would also be useful to compare the AMV indices used here to the AMO itself, since the AMV indices that are a target of this paper looks a lot like the AMO still - eg. negative from 1900 to 1930, positive from 1930 to 1960, negative again from 1960 to 2000, and positive afterwards. Indeed, it isn't clear what has been accomplished by in theory removing the forced signal.

AMO and AMV represent the same thing, a mode of multidecadal variability that represents internal ocean variability. The terminology has moved in the recent years because of some criticism concerning the name AMO, which seems to imply that this mode represents an oscillation, *i.e.*, an harmonic preferential frequency (in the 50-70 yrs band). Given the short time frame of the instrumental observation, it is difficult to robustly state this, which motivated a later re-branding of the mode as Atlantic Multidecadal variability (AMV). For this reason, AMO is much less used in the current and recent literature, and has been superseded by the term AMV.

Removing the forced signal is a key aspect of all the AMV/AMO definitions, and is certainly not a novel aspect of this study. The role of the external forcings on North Atlantic multidecadal variability has been addressed by considering the NASST index, which is indeed compared with the AMV in Figure 6. We have clarified all of this in the revised manuscript to avoid any misunderstanding.

2. The study starts with a very large number of proxy data, which is reduced to approximately ~15% of the initial total. Worryingly, there seems to be little that explains the ultimate importance of these proxies - the majority are from Asia, not Europe or eastern North America, as one might expect.

This reduction of the number of proxy records is a consequence of our objective approach, as we prefer to avoid subjective a priori assumptions. Regarding the regions where proxies are selected, the recent modelling studies by Ruprich-Robert et al. (2017, 2021) clearly show that the AMV climatic signature is not necessarily restricted to the North Atlantic, as the AMV can excite important inter-basin teleconnections.

A nice example of this matter of fact is provided in Fig. 2a and b of Ruprich-Robert et al. (2017) which we show below, highlighting the atmospheric temperature response to an imposed AMV forcing at the ocean surface.

Fig. R3: From Ruprich-Robert et al. (2017) figures 2an and 4ab, representing the temperature differences between the 10-yr average of the ensemble sensitivity simulations showing the response to positive AMV signal for the June to September (JJAS) season in top and December to March (DJFM) in the bottom. Results from (left) CM2.1 and (right) CESM1 models. Stippling indicates regions that are below the 95% confidence level of statistical significance according to a two-sided Student's *t* test. For further details, please refer to Ruprich-Robert et al. (2017).

It clearly shows that Europe is not necessarily the region most affected by the AMV, with Central Asia exhibiting equally strong or even stronger signals across models. The same is true for precipitation (Their Fig. 2e and f) and this is due to dynamical changes in the atmosphere, represented by global and complex changes in sea-level pressure fields (their Fig. 2c and d). Nevertheless, this result might be model-dependent, and not necessarily apply to the real world. Indeed, the realism of air-sea interactions in current models is still partly questioned (Smith et al. 2020), and the exact response of the climate system to the AMV is possibly slightly different to the one in Ruprich-Robert et al (2021). Interestingly, the correlation patterns between the AMV and the global fields of surface temperature and precipitation derived from observations (Figure 1) also indicate that Europe is not necessarily the region with the stronger AMV climatic imprints. This is why we prefer to use the objective approach of selecting only those proxies whose correlation with the observed AMV is significant at least at the 95% confidence level.

Adjacent or co-located proxy records sometimes enter the reconstruction model and sometimes don't. For instance, MOR2JU is part of the reconstruction, but not the co-located MOR3JU and MOR4JU?

Following on from our explanation, our method is only selecting the proxy records that are significantly correlated (at the 95% confidence level) with the AMV over their overlapping period. If not, this means that the considered proxy might not record AMV signature on climate, which can be due to a number of reasons, among which we can have seasonal biases, strong noise, errors in the temporal calibration and the fact that very local environmental conditions can degrade the climatic signals in certain proxies.

Fig. R4: Timeseries of the proxy records labelled “Asia.MOR1JU”, “Asia.MOR2JU”, “Asia.MOR3JU” and “Asia.MOR4JU”.

	Asia.MOR1JU	Asia.MOR2JU	Asia.MOR3JU	Asia.MOR4JU
Asia.MOR1JU	-----	0.5	0.39	0.61
Asia.MOR2JU	0.5	-----	0.82	0.63
Asia.MOR3JU	0.39	0.82	-----	0.56
Asia.MOR4JU	0.61	0.63	0.56	-----

Tab. R1: Correlation table of “Asia.MOR1JU”, “Asia.MOR2JU”, “Asia.MOR3JU” and “Asia.MOR4JU” for the period 1870-1987

Fig. R4 and Tab. R1 illustrate that these 4 particular proxies from Asia, although from close locations and of similar variations, do also have significant differences that can come for the reasons mentioned in the preceding paragraph. It can notably act on the significance of the correlation between the proxy records and the AMV indices, for the different timeframes tested for building statistical models. Since we maximized validation scores among these different statistical models, this implicate that adding one of the two proxies “Asia.MOR2JU” or “Asia.MOR3JU” would lead to lower scores obtained with our model validation process. They indeed might have been used in some of the statistical models tested, but they appeared to not be part of the best one in terms of validation scores. We thus believe that there are no statistically reasonable reasons to add these proxy records in the reconstruction.

(Disturbingly, the authors reveal that a single record from Pakistan MOR1JU weighted heavily in the reconstruction, so they removed it!)

The removal of MOR1JU is not changing the results much: it was included in the preceding review, where similar results were found for the response to external forcings and early warning signal tests. But we believe that to increase the robustness of our results, it is important to avoid a reconstruction where one single proxy has too much weight. It can indeed be seen on Fig. R3 that “Asia.MOR1JU” diverges significantly from the three others before 1930 C.E. Although it was mentioned before that these proxy records do not have the exact same variations, they are at least expected to have similarities. Here, the clear divergence of “Asia.MOR1JU” with the three others during the early part of the historical period might be indicating that “Asia.MOR1JU” does have large uncertainties. We believe that, considering here the use and testing of hundreds of proxies, the occurrence of an abnormally large weight for a single proxy, that in addition has poor consistency with its neighbors, can be expected to occur by chance.

Why would the Maiana coral record from the western Pacific enter into a reconstruction of North Atlantic SSTs? In California, why would the Fish Creek Trail record from Jeffrey Pine enter the model but no other chronologies from California or the western United States?

As stated before, this might be related to the previously mentioned inter-basin teleconnections, and to the local factors that can degrade the quality of climate signals in the different records. We believe it is key to avoid any a priori hypothesis when leading AMV reconstruction given the complex picture of its climate imprints highlighted in recent literature (Ruprich-Robert et al. 2017, 2021). There is of course also the possibility that some proxies correlate significantly with the AMV just by chance, contributing to the reconstruction with a spurious signal. The weights of those proxies, however, are expected to be small as they should be penalised in the cross-validation process. This caveat is now mentioned in the text.

The final list of proxies looks like a random collection but with a weight toward interior Asia, with some exceptions, raising concerns about whether using such a large predictor dataset results in a random selection of proxies. Even in Asia, the sign of the relationship varies between proxies. Indeed, the heavy RF weights associated with sites quite distant from the Atlantic itself raise questions about the proxy inclusion - why wouldn't sites under the more direct influence of the Atlantic show the most importance?

As mentioned in the previous responses there is growing literature reporting very remote AMV impacts, including over interior Asia. The seasonality recorded by the nearby proxies might be a decisive factor explaining their lack of significant correlation with the observed AMV, which is a very solid reason to leave them out of the reconstruction.

The correlations between AMV and proxies reported in Extended Data Table 3 are also quite low (e.g.), especially considering the smoothness of the predictand AMV series. Were the p-values adjusted for reduced degrees of freedom? The authors spend a good part of the manuscript (L186 to L219) in post hoc justification of these unexpected proxy selections.

For the calculation of these correlations, no smoothing is applied to the AMV timeseries, as explained in the second part of our answer to Comment 1, and as it can be seen in Fig. 1 in the manuscript. Since it was not clearly explained, we now discuss with more details this aspect in our manuscript.

According to our approach, they are, among the proxies of the large initial database we start from (named P2k+ in the main text), the most correlated records with the AMV index we reconstruct. We also think that the justification of the proxy records selection is important, and it is even asked to be more detailed on some aspects by Reviewer 2. Indeed, since the most correlated proxies with the AMV (i.e. the proxies we used for the reconstruction) do not necessarily appear in North Atlantic bordering regions (e.g. Asia, western North America...), we believe that referring to literature supporting the inclusion of these proxy records is essential.

2. The authors say the following: 'An important preprocessing step before selecting the proxies is to remove in each proxy record from P2k+ (see Methods) the forced variability of NASST, using linear regression starting in 1870' but nowhere in the methods can I find a description of how this was done. Are the authors suggesting they removed the annual forced NASST signal from local proxies? This seems problematic, as one would expect the local proxy signal to include the direct forcing (especially for temperature) independent of what the Atlantic is doing.

We did not remove the NASST signal in the proxy records, but, as mentioned in the Methods section, we did remove the individual covariations (by linear regression) of each proxy with the forced NASST forced component as estimated by signal-to-noise maximizing EOF from historical CMIP5 models' simulations (Ting et al. 2012). This is crucial since AMV variations and external forcing do exhibit some covariances (e.g. Booth et al. 2012), and therefore, some proxies can end up with significant correlation with the NASST just because they respond to the external forcing, and not directly to the AMV. To avoid such a strong issue, which might have seriously affected former attempt to reconstruct the AMV, we do remove this external forcing signal from the proxy records, in order to focus on the imprints of the AMV, seen as an internal mode of variability, as clarified in recent literature (Ruprich-Robert et al. 2021).

Thus our approach is aiming at disentangling the signature from external forcing that can affect the climate in the location considered and the signal from the (internal) AMV, whose signature is strongly impacting atmospheric dynamics and as shown in modelling studies.

We have further insisted on this aspect in the manuscript, since it is crucial but certainly not explained well in the first version of the paper. We thus thank the reviewer for highlighting this to us.

3. Were the CE scores here benchmarked against red noise? As Wahl and Smerdon 2012 suggest, when degrees of freedom are low and there is the potential for strong red noise, a null test of the CE scores may be warranted. CE is low although in places significant here, but given that a smoothed AMV has relatively few degrees of freedom and the validation period is quite short, an evaluation of whether $CE > 0$ is really the cutoff for significance in this metric would be useful.

This test has been made for the new version of the manuscript. CE scores for the reconstruction are compared to those obtained using simulated red-noise processes as predictors in a new supplementary figure and this aspect is discussed in the main text.

4. It isn't clear why the study selects marine proxies for an ad hoc validation - confusingly, a validation of AMV uses corals from the central Pacific and marine records from the southern hemisphere - it is difficult to see, particularly given the level of smoothing, how this proves and secures validation.

It is always a difficult task to validate paleo-climatic reconstruction notably because of the poor amount (as compared to e.g. present-data observations) of independent data available. Nevertheless, in this paper we have done our best to go beyond existing validation methods, by considering 2 additional validation approaches, on top of the classical calibration/validation model evaluations included in the reconstruction method. Those two additional are i) the use of independent records, i.e. records that have not been used at all in the reconstruction, and ii) the use of pseudo-proxy approach based on models. This comment from the reviewer is concerning the first approach we proposed. We agree with the reviewer that this is not a perfect validation that the independent marine records are providing because the data have quite a low temporal resolution. Nevertheless, we have to work with this shortcoming from the data, and the validation proposed is using statistical tests that account for the smoothness of the data in their computation of the correlation test (similarly to McCarthy et al. 2015 and Michel et al. 2020). In that respect, we believe we have "objectively" (at least statistically speaking) accounted for this shortcoming in the presentation of our results.

The fact that the reconstructed AMV index is statistically significantly correlated with corals from the Central Pacific does not constitute an issue from our point of view. On the contrary, a recent modelling study that looked into teleconnection pattern of the AMV (e.g. Ruprich-Robert et al. 2021) clearly showed that the AMV has a strong impact on the Pacific Ocean, as also highlighted in numerous other modelling studies (Chafik et al. 2016, McGregor et al. 2014).

Thus, although this first validation is not perfect, it is complementary to the two other ones, and provides interesting insights on the consistency between our mostly land-based proxy reconstruction and the marine records. To test the sensitivity to the location of the marine proxies, Fig. 3b shows now a new composite time series only based on the North Atlantic proxies from Fig. 3a (green line), which yields slightly

higher correlations with the AMV reconstruction than the one including marine proxies from all basins.

We have modified the text to improve the explanations of those crucial points, which were not developed enough in the former version of the manuscript. We therefore thank the reviewer for this comment which helped us to see where additional explanations and clarifications were requested in the text to better explain the rigorous methods we have set up to propose our reconstruction.

References:

1. Booth, B. B. B., Dunstone, N. J., Halloran, P. R., Andrews, T. & Bellouin, N. Aerosols implicated as a prime driver of twentieth-century North Atlantic climate variability. *Nature* **484**, 228-232 (2012).
2. Chafik, K., Häkkinen, S., England, M. H., Carton, J. A., Nigam, S., Ruiz-Barradas, A., Hannachi, A., and Miller, L. Global linkages originating from decadal ocean variability in the subpolar North Atlantic. *Geophys. Res. Lett.* **43**,10909–10919 (2016).
3. Knight, J. R., Allan, R. J., Folland, C. K. Vellinga, M. & Mann, M. E. A signature of persistent natural thermohaline circulation cycles in observed climate. *Geophys. Res. Lett.* **32(20)**, L20708 (2005).
4. McCarthy, G. D., Haigh, I. D., Hirshi, J. J.-M., Grist, J. P., Smeed, D. A. Ocean impact on decadal Atlantic climate variability revealed by sea-level observations. *Nature* **521**, 508-512 (2015).
5. McGregor, S., Timmermann, A., Stecker, M., England, M. H., Merrifield, M., Jin, F.-F., and Chikamoto, Y. . Recent Walker circulation strengthening and Pacific cooling amplified by Atlantic circulation. *Nat. Clim. Change*, 4 :888–892 (2014).
6. Michel, S., Swingedouw, D., Ortega, P., Khodri, M., Mignot, J. & Chavent, M. Reconstructing climatic modes of variability from proxy records using ClimIndRec version 1.0. *Geosci. Mod. Dev.* **13**, 841-858 (2020).
7. Ruprich-Robert, Y., Msadek, R., Castruccio, F., Yeager, S., Delworth, T. & Danabasoglu, G. Assessing the climate impacts of the Observed Atlantic Multidecadal Variability Using the GFDL CM2.1 and NCAR CESM1 Global Coupled Models. *J. Clim.* **30(8)**, 2785-2810 (2017).
8. Ruprich-Robert, Y., Moreno-Chamarro, E., Levine, X., Bellucci, A., Cassou, C., Castruccio, F., Davini, P., Eade, R., Gastineau, G., Harmanson, L., Hodson, D., Lohmann, K., Lopez-Parages, J., Monerie, P.-A., Nicolì, D., Qasmi, S., Roberts, C. D., Sanchez-Gomez, E., Danabasoglu, G., Dunstone, N., Martin-Rey, M., Msadek, R., Robson, J., Smith, D. & Tourigny, E. Impacts of Atlantic multidecadal variability on the tropical Pacific: a multi-model study. *NPJ Clim. and Atmos. Sci.* **4**, 33 (2021).

9. Smith, D. M., Scaife, A. A., Eade, R., Athanasiadis, P., Bellucci, A., Bethke, I., Bilbao, R., Borchert, L. F., Caron, L.-P., Counillon, F., Danabasoglu, G., Delworth, T., Doblas-Reyes, F. J., Dunstone, N. J., Estella-Perez, V., Flavoni, S., Hermanson, L., Keenlyside, N., Kharin, V., Kimoto, M., Merryfield, W. J., Mignot, J., Mochizuki, T., Modali, K., Monerie, P.-A., Müller, W. A., Nicolí, D., Ortega, P., Pankatz, K., Pohlmann, H., Robson, J., Ruggieri, P., Sospedra-Alfonso, R., Swingedouw, D., Wang, Y., Wild, S., Yeager, S., Yang, X., and Zhang, L. North Atlantic climate far more predictable than models imply. *Nature* **583**, 796–800 (2020).
10. Ting, M., Kushnir, Y., Saeger, R. & Cuihua, L. Forced and internal twentieth-century SST trends in the North Atlantic. *J. Clim.* **22**, 1469-1481 (2009).
11. Zhang, R., Sutton, R., Danabasoglu, G., Kwon, Y.-O., Marsh, R., Yeager, S. G., Amrhein, D. E. & Little, C. M. A review of the role of Atlantic meridional overturning circulation in Atlantic multidecadal variability and associated climate impacts. *Rev. Geophys.* **57**, 316-375 (2019).

Reviewers' Comments:

Reviewer #2:

Remarks to the Author:

I am satisfied with the authors' response to my initial comments and their edits to the original manuscript. However, I advise the authors and publishing team to take note of the equations and variables, as they do not appear to compile correctly in the version at hand. I therefore cannot comment on the veracity of any equations and must only assume their correctness from the previous version.

Reviewer #4:

Remarks to the Author:

Review of

Early warning for a climate tipping point in a model-tested AMV reconstruction since 850 C.E.

Overview:

The authors reconstruct Atlantic Multidecadal Variability (AMV) using paleoclimate proxy records, and assess the dominant spectral characteristics of the AMV, along with its potential for being an early warning climate tipping point. The paper is interesting and relevant. I appreciate that the authors took care to attempt to objectively assess the proxies, address concerns with non-stationarities, validate the reconstruction, and use red noise to determine if red noise would produce a similar result – these are all great tests that I wish were more widely used.

My experience is in analysis of AOGCM hosing experiments, paleo modeling, paleoclimate field reconstructions, and model-paleo comparisons, so I will generally limit my comments to these areas in the manuscript (for example, I have no experience with early-warning signal analysis, nor do I have experience with random forest reconstruction methods).

Main Comments/Concerns:

1) The title of the paper 'Early warning for a climate tipping point in a model-tested AMV reconstruction since 850 CE' isn't really what the bulk of the paper is about, nor is it in my opinion what the analysis shows (most of the paper is generally sound and interesting, but focused on a new AMV reconstruction that emphasizes internal variability, and the spectral characteristics of this AMV reconstruction, not on AMOC). Specifically, the authors reconstruct AMV, but the AMOC has only been shown to be correlated with AMV on decadal timescales in the HadGEM3-GC2 (according to the Methods section of the referenced Boers, 2021 paper). I am concerned that uncertainty is layered on uncertainty here- there is noise in the AMV reconstruction, and the AMV reconstruction is then correlated with AMOC- how much actual AMOC signal is reconstructed, and how reliably can we claim the AMOC is reaching a tipping point? There are quite a few steps of separation between the reconstructed AMV and the actual AMOC strength, which is never shown or assessed in this paper.

2) I think the idea of reconstructing an AMV index focused on internal variability is interesting and valuable, but the entire reconstruction of an AMV index that focuses on internal variability is based on the idea that the local N Atlantic anthropogenic aerosol forcing dataset used in the CMIP5 historical runs is 'correct', and that the CMIP5 multi-model mean can be used to reliably remove the local forced signal in the reconstruction process. This method (using CMIP5 MMM to remove forced signal) is a sensible choice, but introduces a whole new potential set of biases that could be imposed on the reconstruction that are basically not discussed, beyond one sentence in the manuscript. For example, see Fyfe et al. (PNAS, 2021: <https://doi.org/10.1073/pnas.2016549118>), who show: 'significant differences in simulated global surface air temperature due to volcanic aerosol forcing in the second

half of the 19th century and in the early 21st century. The latter arise from small-to-moderate eruptions incorporated in CMIP6 simulations but not in CMIP5 simulations. We also find significant differences in global surface air temperature and Arctic sea ice area due to anthropogenic aerosol forcing in the second half of the 20th century and early 21st century. These differences are as large as those obtained in different versions of an Earth System Model employing identical forcings.' I would suggest the authors address this issue more directly, and include more of a discussion of this potential issue, and how it could be overcome or impact the results. I would suggest using the CESM LME pseudoproxy reconstruction to address this issue, but even if the pseudoproxy reconstructions have the CMIP5 MMM signal removed, I assume similar aerosol/GHG forcing is used in the CESM LME, so this wouldn't really get around the problem. Are the results sensitive to using the CMIP6 MMM signal removal?

3) As other reviewers have noted, the results contain quite a bit of Methods, and also present a lot of what looks like to me to be 'beside the point' sensitivity testing. For example, I think the CESM pseudoproxy analysis is an important sensitivity/robustness test, but is not really relevant to the main findings of the paper. I would suggest moving this entire analysis (and associated methods/results/figures) to the supplement. This would allow the paper to focus on the main findings of the AMV reconstruction. I would also suggest moving a lot of the sensitivity testing to the Methods or supplement.

4) Also, as other reviewers have noted, the removal on Southern Hemisphere proxies seems strangely subjective to me. A strength of the paper is its claim that proxy information was objectively chosen, regardless of location/proximity to the North Atlantic (and the authors do a good job showing that there are locations in Asia, etc. that are strongly correlated/related to AMV), then the paper goes on to exclude southern hemisphere proxies. If indeed the S Hemisphere proxies don't relate to variability in the N Atlantic, then let the method itself down-weight/exclude that proxy information, and don't subjectively remove the proxy data. I understand another reviewer didn't like the inclusion of S Hemisphere proxy data, but it seems to undercut the paper's methods to subjectively exclude the data.

5) I found it nearly impossible to review most of the Methods because the symbols did not render correctly- most of AMV etc symbols were replaced with '?', so I could not follow/review this section.

6) The often awkward and/or extraneous wording in the paper makes it difficult to read. See my specific line comments below. This is a significant issue for me because I expect a Nature Communications paper to be well written and easy to follow.

Specific Comments:

Line 19: what are 'crucial' impacts?

Line 20: suggest a word other than 'unconsensual'

Lines 25-27: 'The best performing reconstruction, when verified both against climate model outputs and independent proxy records, is obtained using the random forest method.' – what random forest method? This seems to come out of nowhere- can the authors first introduce how this is relevant/what they did?

Line 27: What is 'It' at the start of the sentence? Please be more specific- as it is written, 'it' could be the method, not the AMV

Line 35: suggest 'effectively amplifying or dampening' in place of 'effectively contributing to amplify or dampen'

Line 43: 'global and regional impacts it will have in a context of climate change' suggest 'the' instead of 'a'

Line 44: given that this journal is intended for a broad scientific audience, I suggest explaining spectral (perhaps 'dominant timescales of variation?') before using the term

Line 70: 'allowing to avoid'- remove 'allowing'

Line 70 'This is the type of definition' – suggest 'this study uses xx definition' – I was confused – which definition? Please just state what it is.

Line 80: 'unveil'- reveal?

Lines 91-96: Mann et al. show this in 2021, but Parsons et al. (2020, GRL) also show that several CMIP6 piControl simulations show robust internal/unforced, global-mean, multidecadal temperature variability, which is strongly associated with North Atlantic variability (so not essentially driven by volcanic forcing).

Parsons, L. A., Brennan, M. K., Wills, R. C. J., & Proistosescu, C. (2020). Magnitudes and spatial patterns of interdecadal temperature variability in CMIP6. *Geophysical Research Letters*, 47, e2019GL086588. <https://doi.org/10.1029/2019GL086588>

Lines 110-112: 'but the existence of multi-decadal variations in preindustrial control simulations of AOGCMs, which do not include external forcing, still indicates physical arguments in favor of a role of internal variability related to ocean circulation in producing variations of the AMV.'
Again, suggest Parsons et al., 2020, GRL who show this for many CMIP6 models.

Line 121: 'implying time series' suggest instead 'implied from a time series'

Lines 135-136: 'but also strongly affect' - suggest 'are associated with'- causality is not shown here

Lines 149-155: Another recent study, by Singh et al. (2018, *Clim. Past*), used paleo data assimilation to reconstruct AMV:

Singh, H. K. A., G. J. Hakim, R. Tardif, J. Emile-Geay, and D. C. Noone (2018), Insights into Atlantic multidecadal variability using the Last Millennium Reanalysis framework, *Clim. Past*, 14(2), 157-174, doi:10.5194/cp-14-157-2018.

Lines 187-195: See my main comment above- does this method imprint CMIP5 bias onto reconstruction or only partly remove the forced signal if the CMIP5 forcing dataset isn't exactly right? For example, what if the forcing dataset in CMIP5 models is spatially imposed incorrectly for anthropogenic aerosols, thereby removing the wrong signal, and imprinting a non-climate bias signal onto the reconstructions? Lines 197-199 do acknowledge the assumption that the CMIP5 models are capturing the forced response, but I think more discussion (here and/or later in Discussion/Conclusion/Methods) or potential testing is warranted.

Line 203: what makes CE 'classical'?

Line 222: confusing wording 'It consists in increasingly reconstructing the AMVF index over time'

Lines 224-228: thank you for doing this test- adds to robustness/believability of results

Line 231: '(avg(CE)=-0.08' - this is confusing- this wording makes it sound like the CE of the real proxies is -0.08. Suggest just including the real score.

Lines 239-243: would be helpful to note what the reconstruction is tested against, here, and/or in figure caption- I searched briefly and did not see it, which means the average lazy reader wouldn't find it

Line 269: 'the method is also selecting' - awkward wording- suggest something like 'the method selects'

Line 273: 'to be communicating'- suggest 'to communicate'

Line 276: 'both' - implies two ideas will be listed, but only one is listed (?)

Line 286: 'constrain'- suggest 'constraint'

Lines 288-289: 'to perform a reconstruction with potentially a few spurious predictors.' - awkward wording, suggest changing

Line 295-300: Glad the authors used withheld data to test their reconstruction.

Lines 311-312: 'This validation is important and notably witnesses for a satisfying' - what does this mean?

Lines 320-321: I think it is just "Last Millennium Ensemble", not LMLE (see Otto-Bliesner et al documentation paper for how they refer to the CESM LME)

Lines 315-364: This is a great validation section, but this all seems like Supplemental material, or Methods that are testing the main results - I would move this to SI. It is really interesting, but in my opinion not relevant to the main result other than to help show the reader that the reconstruction is robust.

Lines 366-372: Another reconstruction? this makes sense, but be sure to introduce earlier so not a 'surprise' to the reader (or did I miss this being introduced earlier? Lines 74-76 seem to suggest there are 3 instrumental era definitions of AMV that will be reconstructed, but where is this NASST reconstruction introduced?)

Lines 378-381: 'this suggests that about 40% of the variability...the rest being related to external forcing' - What if there is noise in the reconstructions unrelated to AMV or external forcing? I'm not sure about the assumption that any signal not shared in the two reconstructions is just due to forcing.

Lines 419-420: 'They are also questioning...' suggest 'There are also'

Lines 429-430: 'The broader range for the AMV timescale of variability we find is also produced in control simulations with fixed external forcings from some climate models' - and in global variability associated with North Atlantic variability in several CMIP6 models (Parsons et al., 2020, GRL, see above ref)

Lines 441-444: 'they also have timescales of variations on lower multidecadal frequencies (e.g., 20 to 50-year). Thus these results...' These are great points, but I would like to highlight that Singh et al. show similar results for AMV based on paleo data assimilation reconstructions (2018, Clim. Past, see above ref)

Lines 446-504: I find this section quite interesting, but have no experience with these EWS detection methods, so have not commented. See my main comment above on the tenuous assumption that 'reconstructed AMV' = 'real AMV' = 'real AMOC' because 'simulated AMV' is similar to 'simulated AMOC'

in one Had model simulation – there are quite a few steps of separation between the reconstructed AMV and the actual AMOC strength, which is never shown or assessed in this paper.

Lines 496-497: 'AMV time series which by construction only reflects internal variability and thus might well reflect AMOC pulses' – is this ever assessed/shown?

Line 446: 'Early warning signal for an approaching tipping point' – The paper seems to end on a Results section, with a brief climate disaster warning based on the inferences made from the results. Can the authors mention anywhere the potential weaknesses/assumptions made throughout (eg forced signal in CMIP models, AMOC=AMV)? Or mention where we go from here as a scientific community? Off the top of my head, these results also suggest a major need to better understand and observe the actual AMOC over a long enough time period to assess the hypothesized multidecadal relationship with AMV, for example, instead of relying on model simulations.

Line 795: Why this arbitrary cutoff at the equator? Clearly proxies in Asia can show correlations with the AMV region, so why not the tropical South Atlantic or similar regions? The rebuttal figure R3 from the authors does seem to suggest there are several locations in the western Pacific/SPCZ, Indian ocean, and South America that would show significant relationships with AMV.

Lines 803-804: 'Interestingly, this decision did not majorly change the final reconstruction.' can the authors show this somewhere, present results from a statistical comparison? Or at least admit it's qualitatively similar?

Lines 830-end: unable to review Methods due to lack of ability to read symbols, etc.

Lines 1023-1029: What data do boxplots show in the figures? I could not determine this from the figure caption, or from this Methods section.

Figure 3: this is a great test of the reconstruction, but seems to be to be a Methods/Supplement figure for a Nature Communications paper

Figure 4: how are the boxplots made? I understand the statistics they show, but I don't understand the individual data points that go into the boxplots- what is the distribution showing? Is the time series divided up within each reconstruction in the CESM runs 1-13? Or what data go into the boxplots (other than the model-derived AMV and the pseudoproxy AMV in model space, I get this) Also, again, this Figure seems like a Supplemental/Methods type of analysis, not the main result

Figure 5: I don't understand what the purple lines mean, or why they are lines and not just points (?) is there a connection between the CESM LME ensemble members the reader is supposed to follow, ensemble 1-13?

And again, see my comment from Figure 4- what data go into the boxplots?

Figure 6: 'Composite series are performed for 31 years, with the 11th year being the year of the eruptions. Each individual response is centered to its values 10 years before the eruption (from N-10 to N-1)' – this explanation was confusing to me. Isn't year 0 the year of the eruption, and year negative 10 the 10th or 11th year before the eruption?

'centered to its value'- does this mean it's normalized or the mean is removed from this time period?

Reviewer #2 (Remarks to the Author):

I am satisfied with the authors' response to my initial comments and their edits to the original manuscript. However, I advise the authors and publishing team to take note of the equations and variables, as they do not appear to compile correctly in the version at hand. I therefore cannot comment on the veracity of any equations and must only assume their correctness from the previous version.

We thank the reviewer for the overall review of our manuscript that helped a lot to improve it. For the equations, we are sorry to see this happened after the Word to PDF conversion from the journal portal. Following submissions will be made by PDF files directly, where we will check cautiously if the equations have been correctly converted.

Reviewer #4 (Remarks to the Author):

Review of

Early warning for a climate tipping point in a model-tested AMV reconstruction since 850 C.E.

Overview:

The authors reconstruct Atlantic Multidecadal Variability (AMV) using paleoclimate proxy records, and assess the dominant spectral characteristics of the AMV, along with its potential for being an early warning climate tipping point. The paper is interesting and relevant. I appreciate that the authors took care to attempt to objectively assess the proxies, address concerns with non-stationarities, validate the reconstruction, and use red noise to determine if red noise would produce a similar result – these are all great tests that I wish were more widely used.

My experience is in analysis of AOGCM hosing experiments, paleo modeling, paleoclimate field reconstructions, and model-paleo comparisons, so I will generally limit my comments to these areas in the manuscript (for example, I have no experience with early-warning signal analysis, nor do I have experience with random forest reconstruction methods).

Main Comments/Concerns:

1) The title of the paper 'Early warning for a climate tipping point in a model-tested AMV reconstruction since 850 CE' isn't really what the bulk of the paper is about, nor is it in my opinion what the analysis shows (most of the paper is generally sound and interesting, but focused on a new AMV reconstruction that emphasizes internal variability, and the spectral characteristics of this AMV reconstruction, not on AMOC). Specifically, the authors reconstruct AMV, but the AMOC has only been shown to be correlated with AMV on decadal timescales in the HadGEM3-GC2 (according to the Methods section of the referenced Boers, 2021 paper). I am concerned that uncertainty

is layered on uncertainty here- there is noise in the AMV reconstruction, and the AMV reconstruction is then correlated with AMOC- how much actual AMOC signal is reconstructed, and how reliably can we claim the AMOC is reaching a tipping point? There are quite a few steps of separation between the reconstructed AMV and the actual AMOC strength, which is never shown or assessed in this paper.

The link between the AMV and the AMOC has been found in many models. It was first identified by Knight et al. (2005) in HadCM3, a predecessor lower resolution version of HadGEM3-GC2, and since then has been confirmed in many other climate models: CMIP3 models (Ba et al., 2014), CESM1 (Kim et al., 2018), CMIP5 models (Muir and Fedorov, 2015; Yan et al 2018), IPSL-CM5A (Persechino et al 2013; Gastineau et al. 2016, among others). We would also like to clarify that in its Method section Boers (2021) referred to the links between the AMOC and other SST and SSS based indices that are different from the AMV. Those linkages have indeed only been confirmed with HadGEM3-GC2 by Jackson and Wood (2018), contrary to the wide range of validation for the AMV-AMOC relationship (references above).

We think that within the large controversy concerning the drivers of observed AMV that we discuss extensively in the introduction, there is a large literature, very well reviewed and discussed by Zhang et al. (2019), who highlighted the key role of the AMOC for the AMV variations, and *vice versa*. Although this link is not necessarily a one-to-one correlation across the whole range of available GCMs, it has also been pointed out that it may be largely underestimated by GCMs (Yan et al. 2018). It is on the basis of this literature that we attempt to reconstruct the AMV and eventually relate it to the AMOC.

To further strengthen the references regarding AMOC/AMV relationships, we also provide additional analysis of the CESM-LME ensemble. The use of CESM-LME is chosen here because historical runs are too short to study multidecadal relationships between time series and also include large anthropogenic forcings which might blur the signal (as found in Frankignoul et al. 2017). In addition to the last millennium AMV_F indices we extracted for the PPE, we also computed AMOC indices from the same simulations of our ensemble. We use one commonly used AMOC index, namely the maximum streamfunction below 500 meters as a maximum between 20°N and 65°N ($AMOC_{max}$). The relationship using other AMOC indices such as the maximum streamfunction below 500 meters at 26.5°N ($AMOC_{26.5}$) and 34°N ($AMOC_{34}$) was also checked and the analysis of these two indices has been put in Supplementary Information.

We first show for each member of the simulation the cross-correlations between the 10-year smoothed $AMOC_{max}$ index and the 10-year smoothed AMV_F index in CESM-LME, provided in Fig. R1.

Figure R1: Cross-correlations between $AMOC_{max}$ and AMV_F in 12 members from CESM-LME. a-l) Members number 2 to 13, respectively. $AMOC_{max}$ leads (resp. lags) AMV_F for negative (resp. positive) time lags (expressed in years). Significant cross-correlations are indicated by orange ($0.05 < p < 0.1$), red ($0.01 < p < 0.05$) and green ($p < 0.01$) dots. Significance is calculated by accounting for the time series autocorrelations (McCarthy et al. 2015, Michel et al. 2020, Methods). All time series were previously smoothed with a 10-year kernel filter.

The results of the maximum lagged correlations when AMOC is leading, and its significance as calculated for correlations over the entire study (i.e., accounting for time series autocorrelations, McCarthy et al. 2015, Michel et al. 2020) are summarized in Figure R2.

Figure R2: Maximum lagged correlations when AMOC indices lead the AMV_F index in 12 members of CESM-LME. Circles, squares and diamonds correspond to the maximum lagged correlation when $AMOC_{26.5}$, $AMOC_{34}$, and $AMOC_{max}$ respectively lead the AMV_F index. Orange, red and green points indicate 90%, 95%, and 99% confidence levels, respectively. White points indicate no significance at the 90% confidence level. Numbers in each point indicate the time step where the maximum correlation is reached when AMOC indices lead the AMV_F index. All time series are smoothed with a 10-year kernel filter.

Across the members, the maximum cross-correlations is found for the AMOC leading in a range from 4 to 8 years, which are significant at least at the 90% confidence level in 10 out of 12 members. The discrepancies between members may reflect the influence of internal variability (El Niño Southern Oscillation, North Atlantic Oscillation, among others), and/or the manifestation of potentially different long-term drifts.

In terms of the relevance of using EWS statistics on AMV to then hypothesize future AMOC EWS, we further provide another analysis from CESM-LME simulations (Fig. R3). For the 12 members, we computed the EWS approaches from the millennial model simulations for the $AMOC_{max}$ index and the AMV_F index, for 3 different window lengths (WL=200, 300 and 400 years) and the associated Kendall tau statistics (see Methods). It appears that the Kendall tau statistics calculated from the AOGCM simulations are, for the ensemble of simulations, consistent between those obtained for the AMV_F index and those obtained for the $AMOC_{max}$ index. This indicates that although the AMV_F does not completely capture all the variations of the AMOC, it does significantly capture the changes in its memory as measured for EWS detection (Fig. R3). A Supplementary Figure is also added with the same analysis for two other AMOC indices (*i.e.*, $AMOC_{26}$ and $AMOC_{34}$).

Figure R3: Scatterplot of Kendall tau statistics of AR1 coefficients for AMV and AMOC indices in CESM-LME. Kendall tau statistics obtained the AMV_{max} index (ordinates) are plotted against the Kendall tau statistics obtained for the index AMV_F index (abscissas). Plotted numbers indicate the member index of the CESM-LME simulations. Red lines are the ordinary least squares regression lines, and their significance is calculated using a two-tailed Student test.

Fig. R1 has been added as supplementary figures and the corresponding analysis is explained in the Supplement. Likewise, Fig. R2 and R3 have been added and are discussed in the main text to support the results from the study. For the sake of objectivity, we also point out in the text that while CESM-LME provides interesting millennial simulations that are not as accessible for other AOGCMs, our AMOC/AMV

analyses here are based on a single model. The Zenodo repository will be updated with the code and data to produce these additional analyses.

With the above changes, we decided to keep the original title, as we now provide analyses supporting the discussed link between the AMOC and the AMV, and also show that this link holds for the relevant metric used to define the EWS. We believe that the study is now more in balance with the title and the abstract.

2) I think the idea of reconstructing an AMV index focused on internal variability is interesting and valuable, but the entire reconstruction of an AMV index that focuses on internal variability is based on the idea that the local N Atlantic anthropogenic aerosol forcing dataset used in the CMIP5 historical runs is 'correct', and that the CMIP5 multi-model mean can be used to reliably remove the local forced signal in the reconstruction process. This method (using CMIP5 MMM to remove forced signal) is a sensible choice, but introduces a whole new potential set of biases that could be imposed on the reconstruction that are basically not discussed, beyond one sentence in the manuscript. For example, see Fyfe et al. (PNAS, 2021: <https://doi.org/10.1073/pnas.2016549118>), who show: 'significant differences in simulated global surface air temperature due to volcanic aerosol forcing in the second half of the 19th century and in the early 21st century. The latter arise from small-to-moderate eruptions incorporated in CMIP6 simulations but not in CMIP5 simulations. We also find significant differences in global surface air temperature and Arctic sea ice area due to anthropogenic aerosol forcing in the second half of the 20th century and early 21st century. These differences are as large as those obtained in different versions of an Earth System Model employing identical forcings.' I would suggest the authors address this issue more directly, and include more of a discussion of this potential issue, and how it could be overcome or impact the results. I would suggest using the CESM LME pseudoproxy reconstruction to address this issue, but even if the pseudoproxy reconstructions have the CMIP5 MMM signal removed, I assume similar aerosol/GHG forcing is used in the CESM LME, so this wouldn't really get around the problem. Are the results sensitive to using the CMIP6 MMM signal removal?

We initially decided not to use the ensemble of CMIP6 simulations, due to the recent research pointing out that they may overestimate the North Atlantic response to aerosol effects compared to CMIP5 simulations for the historical period (Menary et al. 2020). Our decision to use CMIP5 forcings for removing the forced response was also motivated by the fact that, and as mentioned by the reviewer, the pseudo-proxy analysis is based on CESM-LME, which uses CMIP5 forcings, making thus the whole study more consistent.

However, we agree with the reviewer that a large uncertainty exists concerning the radiative forcing of the aerosols, which is expected to affect the realism of our reconstruction. For this reason, to assess to what extent this uncertainty can affect the main conclusions of our study, we have repeated the reconstruction procedure using in

this case a 30-model CMIP6 ensemble (see Table R1) instead to estimate the externally forced signal. The best reconstruction is obtained by applying the principal component regression method to the reconstruction of the AMV_F index over the period 850-1990 with a slightly higher mean score than the best reconstruction based on CMIP5 forcing (0.242 vs. 0.234). The new reconstruction obtained when using CMIP6 models to remove external forcing is significantly correlated at the 99% confidence level with our reconstruction based on CMIP5 forcings ($r=0.85$ for 10-year smoothed time series, $r=0.76$ for annually-resolved time series) as shown in Fig. R3.

Figure R3: Best AMV reconstruction using two different set of models (CMIP5 in light blue and CMIP6 in purple) to remove the external forcing on the historical period.

Model name	Research center (Country)
ACCESS-CM2	ACCESS (Australia)
ACCESS-ESM1-5	ACCESS (Australia)
BCC-CM2-MR	BCC (China)
BCC-ESM1	BCC (China)
CAMS-CSM1-0	CAMS (China)
CESM2	NCAR (USA)
CESM2-FV2	NCAR (USA)
CESM2-WACCM	NCAR (USA)
CIesm	THU (China)
CMCC-CM2-HR4	CMCC (Italy)
CMCC-CM2-SR5	CMCC (Italy)
CMCC-ESM2	CMCC (Italy)
CanESM5	CCCma (Canada)
FGOALS-f3-L	CAS (China)
FGOALS-g3	CAS (China)
FIO-ESM-2-0	QLNM (China)
GFDL-CM4	GFDL (UK)
GFDL-ESM4	GFDL (UK)
GISS-E2-1-H	NASA GISS (USA)
IPSL-CM6A-LR	IPSL (France)

MCM-UA-1-0	UA (USA)
MIROC6	JAMSTEC (Japan)
MPI-ESM1-2-HAM	MPI (Germany)
MPI-ESM1-2-HR	MPI (Germany)
MPI-ESM1-2-LR	MPI (Germany)
MRI-ESM2-0	MRI (Japan)
NESM3	NUIST (China)
NorCPM1	NCC (Norway)
NorESM2-LM	NCC (Norway)
NorESM2-MM	NCC (Norway)

Table R1: List of the CMIP6 models used for the sensitivity test of the reconstruction of the AMV

Most importantly, using the CMIP6 forcings does not lead to different scientific conclusions as shown in Fig. R4-R6, where the main scientific results from Fig. 2,6,7 as numbered in the previous version of the manuscript are identically reproduced.

Given the aforementioned limitations of the CMIP6 forcings (cf. Menary et al. 2020), we keep the use of the CMIP5 forcings to compute the AMV reconstruction in the main manuscript, and have added the new figures and Tables to the Supplementary Information section, which are now discussed in the context of an additional sensitivity test.

Figure R4. Same as Fig. 3 (from the new version of the manuscript) but with CMIP6 generation used to remove external forcing from proxy records to produce the reconstruction (Fig. R3).

Figure R5. Same as Fig. 4 (from the new version of the manuscript) but with CMIP6 generation used to remove external forcing from proxy records to produce the reconstruction (Fig. R3).

Figure R6. Same as Fig. 6 (from the new version of the manuscript) but with CMIP6 generation used to remove external forcing from proxy records to produce the reconstruction (Fig. R3).

3) As other reviewers have noted, the results contain quite a bit of Methods, and also present a lot of what looks like to me to be 'beside the point' sensitivity testing. For example, I think the CESM pseudoproxy analysis is an important sensitivity/robustness test, but is not really relevant to the main findings of the paper. I would suggest moving this entire analysis (and associated methods/results/figures) to the supplement. This would allow the paper to focus on the main findings of the AMV reconstruction. I would also suggest moving a lot of the sensitivity testing to the Methods or supplement.

We have moved most of this discussion on the pseudo-proxy analyses to the supplement as suggested by the reviewer. In the main text, we have kept a synthesis paragraph of these pseudo-proxy analyses in order to explain their added value and the fact that the results they bring reinforce the validity and robustness of our reconstruction. Based on the advice of the reviewer, the two figures associated with this section have been also moved to the Supplement (as Supplementary Figures 10 and 11). In addition, paragraphs interpreting the results and all the technical details are now provided in the Methods and Supplementary sections.

4) Also, as other reviewers have noted, the removal on Southern Hemisphere proxies seems strangely subjective to me. A strength of the paper is its claim that proxy information was objectively chosen, regardless of location/proximity to the North Atlantic (and the authors do a good job showing that there are locations in Asia, etc. that are strongly correlated/related to AMV), then the paper goes on to exclude southern hemisphere proxies. If indeed the S Hemisphere proxies don't relate to variability in the N Atlantic, then let the method itself down-weight/exclude that proxy information, and don't subjectively remove the proxy data. I understand another reviewer didn't like the inclusion of S Hemisphere proxy data, but it seems to undercut the paper's methods to subjectively exclude the data.

We understand the reviewers' concerns about the exclusion of these proxies. When we previously addressed the other reviewer request, we noticed that adding Southern Hemisphere proxy records significantly lowered the mean score for the best of all the reconstructions (0.19 against 0.23, $p < 0.05$). This best reconstruction, including Southern Hemisphere proxies, is obtained using the Elastic-Net method to reconstruct the AMV_{τ} index for the period 850-1987. Despite the lower mean score, the reconstruction obtained by adding Southern Hemisphere records is very close to the original one, as it is significantly correlated at the 99% confidence level with our AMV reconstruction based on Northern Hemisphere proxy records only ($r = 0.88$ when ten-year filtered, $r = 0.8$ for annual time series), Fig. R7). Southern hemisphere records were included in a former version of the manuscript and the main results and conclusions were similar to the current ones.

Figure R7: Best AMV reconstruction obtained by using NH proxy records only (blue) and NH+SH proxy records (green).

A close inspection reveals that the new reconstruction only includes proxy records from Oceania in the Southern Hemisphere, all with relatively low weights compared with other proxy records from the Northern Hemisphere. The lack of representativity of proxies from other regions like South America and the southern tropical band might be due to the limited availability of proxies from those regions. In the P2k+ records, most of these proxies do not have adequate temporal resolution and/or are not significantly correlated with the closest CRUTS4 (Harris et al. 2020) grid point, which are requisites for proxy records pre-screening (see Methods).

5) I found it nearly impossible to review most of the Methods because the symbols did not render correctly- most of AMV etc symbols were replaced with '?', so I could not follow/review this section.

We sincerely apologize for this inconvenience, which we discovered at the same time as the review reports. It seems to have occurred during the conversion from Word to PDF from the journal system, as equations still appear on the Word version of our manuscript. For the next version, we will submit the PDF files directly and check them carefully before doing so.

6) The often awkward and/or extraneous wording in the paper makes it difficult to read. See my specific line comments below. This is a significant issue for me because I expect a Nature Communications paper to be well written and easy to follow.

We understand this major concern from the reviewer. In addition to the nicely suggested specific comments from the reviewer, we have put further efforts to try to

improve the English language through the whole document. We also reformulate many sentences to make the text easier to follow.

Specific Comments:

Line 19: what are 'crucial' impacts?

We agree this statement is not clear. We replaced "crucial" by "strong" in the new version of the document.

Line 20: suggest a word other than 'unconsensual'

We now rather state that it is "widely debated" instead of using "unconsensual".

Lines 25-27: 'The best performing reconstruction, when verified both against climate model outputs and independent proxy records, is obtained using the random forest method.' – what random forest method? This seems to come out of nowhere- can the authors first introduce how this is relevant/what they did?

We have re-written the abstract where this aspect is better introduced.

Line 27: What is 'It' at the start of the sentence? Please be more specific- as it is written, 'it' could be the method, not the AMV

This sentence was removed in the re-writting process.

Line 35: suggest "effectively amplifying or dampening" in place of 'effectively contributing to amplify or dampen'

Done

Line 43: 'global and regional impacts it will have in a context of climate change' suggest 'the' instead of 'a'

This sentence was removed in the re-writting process.

Line 44: given that this journal is intended for a broad scientific audience, I suggest explaining spectral (perhaps 'dominant timescales of variation?') before using the term

Done

Line 70: 'allowing to avoid'- remove 'allowing'

Done

Line 70 'This is the type of definition' – suggest 'this study uses xx definition' – I was confused – which definition? Please just state what it is.

Done

Line 80: 'unveil'- reveal?

We replaced "unveil" by "reveal"

Lines 91-96: Mann et al. show this in 2021, but Parsons et al. (2020, GRL) also show that several CMIP6 piControl simulations show robust internal/unforced, global-mean, multidecadal temperature variability, which is strongly associated with North Atlantic variability (so not essentially driven by volcanic forcing).

We thank the reviewer for adding this reference which helps to support our results, and to detail better the current AMV bibliography in our introduction.

Parsons, L. A., Brennan, M. K., Wills, R. C. J., & Proistosescu, C. (2020). Magnitudes and spatial patterns of interdecadal temperature variability in CMIP6. *Geophysical Research Letters*, 47, e2019GL086588. <https://doi.org/10.1029/2019GL086588>

Lines 110-112: 'but the existence of multi-decadal variations in preindustrial control simulations of AOGCMs, which do not include external forcing, still indicates physical arguments in favor of a role of internal variability related to ocean circulation in producing variations of the AMV.'

Again, suggest Parsons et al., 2020, GRL who show this for many CMIP6 models.

Done

Line 121: 'implying time series' suggest instead 'implied from a time series'

Done

Lines 135-136: 'but also strongly affect' - suggest 'are associated with'- causality is not shown here

Done

Lines 149-155: Another recent study, by Singh et al. (2018, *Clim. Past*), used paleo data assimilation to reconstruct AMV:

Singh, H. K. A., G. J. Hakim, R. Tardif, J. Emile-Geay, and D. C. Noone (2018), Insights into Atlantic multidecadal variability using the Last Millennium Reanalysis framework, *Clim. Past*, 14(2), 157-174, doi:10.5194/cp-14-157-2018.

This reference has been added in this paragraph.

Lines 187-195: See my main comment above- does this method imprint CMIP5 bias onto reconstruction or only partly remove the forced signal if the CMIP5 forcing dataset isn't exactly right? For example, what if the forcing dataset in CMIP5 models is spatially imposed incorrectly for anthropogenic aerosols, thereby removing the wrong signal, and imprinting a non-climate bias signal onto the reconstructions? Lines 197-199 do acknowledge the assumption that the CMIP5 models are capturing the forced response, but I think more discussion (here and/or later in Discussion/Conclusion/Methods) or potential testing is warranted.

As indicated in our response to main comment 2), we have added the same analysis and figures with CMIP6-based external forcing. As shown in the supplementary, this does not notably affect the reconstructions, nor the main scientific results obtained. See our response to main comment 2.

Line 203: what makes CE 'classical'?

This is indeed wrongly stated. We now define it as "a commonly used metric" and use references to support this statement.

Line 222: confusing wording 'It consists in increasingly reconstructing the AMVF index over time'

We replaced this statement by "This involves progressively reconstructing the AMV_F index over time (one year at a time, starting with the oldest) by constructing a new RF model including the newly available proxies at each time step (see Methods)"

Lines 224-228: thank you for doing this test- adds to robustness/believability of results
We thank the reviewer for appreciating this effort.

Line 231: '(avg(CE)=-0.08' - this is confusing- this wording makes it sound like the CE of the real proxies is -0.08. Suggest just including the real score.

We replaced this wording by "[...] significantly negative (-0.08, p<0.01) [...]"

Lines 239-243: would be helpful to note what the reconstruction is tested against, here, and/or in figure caption- I searched briefly and did not see it, which means the average lazy reader wouldn't find it

We replaced this sentence by "The nested RF reconstruction of the AMV uses a total of 55 proxy records from the Northern Hemisphere (Fig. 2a,b) selected from their correlation with the AMV_F."

Line 269: 'the method is also selecting' - awkward wording- suggest something like 'the method selects'

Replace by "the method selects"

Line 273: 'to be communicating'- suggest 'to communicate'

Done

Line 276: 'both' – implies two ideas will be listed, but only one is listed (?)

"both" have been removed in this sentence.

Line 286: 'constrain'- suggest 'constraint'

Done

Lines 288-289: 'to perform a reconstruction with potentially a few spurious predictors.' – awkward wording, suggest changing

We replaced this sentence by "which implies that some spurious predictors might be used in the reconstruction"

Line 295-300: Glad the authors used withheld data to test their reconstruction.

Lines 311-312: 'This validation is important and notably witnesses for a satisfying' – what does this mean?

We replaced “witnesses for a satisfying” by “indicates a satisfactory”.

Lines 320-321: I think it is just “Last Millennium Ensemble”, not LMLE (see Otto-Bliesner et al documentation paper for how they refer to the CESM LME)

We thank the reviewer for pointing out this issue. We now use the abbreviation “LME” and corrected the full name of the ensemble.

Lines 315-364: This is a great validation section, but this all seems like Supplemental material, or Methods that are testing the main results - I would move this to SI. It is really interesting, but in my opinion not relevant to the main result other than to help show the reader that the reconstruction is robust.

We have synthesized in the main text all the information on the validation procedure, and moved the most technical details to the Methods and Supplementary Information sections.

Lines 366-372: Another reconstruction? this makes sense, but be sure to introduce earlier so not a 'surprise' to the reader (or did I miss this being introduced earlier? Lines 74-76 seem to suggest there are 3 instrumental era definitions of AMV that will be reconstructed, but where is this NASST reconstruction introduced?)

Following the reviewer's suggestion, we now mention this NASST reconstruction and why it is performed just after the AMV reconstruction is discussed in the first section of the results.

Lines 378-381: 'this suggests that about 40% of the variability...the rest being related to external forcing' – What if there is noise in the reconstructions unrelated to AMV or external forcing? I'm not sure about the assumption that any signal not shared in the two reconstructions is just due to forcing.

The reviewer is right, we cannot properly infer whether the unexplained variability comes from the forcing. We decided to remove this sentence which is, however, not central in our study.

Lines 419-420: 'They are also questioning...' suggest 'There are also'

This sentence was removed in the rewriting process

Lines 429-430: 'The broader range for the AMV timescale of variability we find is also produced in control simulations with fixed external forcings from some climate models' – and in global variability associated with North Atlantic variability in several CMIP6 models (Parsons et al., 2020, GRL, see above ref)

This reference is now used to support our results in this section.

Lines 441-444: 'they also have timescales of variations on lower multidecadal frequencies (e.g., 20 to 50-year). Thus these results...' These are great points, but I would like to highlight that Singh et al. show similar results for AMV based on paleo data assimilation reconstructions (2018, Clim. Past, see above ref)

This reference is now provided in the new version of the manuscript

Lines 446-504: I find this section quite interesting, but have no experience with these EWS detection methods, so have not commented. See my main comment above on the tenuous assumption that 'reconstructed AMV' = 'real AMV' = 'real AMOC' because 'simulated AMV' is similar to 'simulated AMOC' in one Had model simulation – there are quite a few steps of separation between the reconstructed AMV and the actual AMOC strength, which is never shown or assessed in this paper.

The reviewer may refer to our response to main comment 1). We now analyse the links between the AMV and AMOC in CESM-LE. In addition to the various references we provide (Muir and Fedorov 2015, Yan et al. 2018, Zhang et al. 2019, references in response to main comment 1), we also show that the EWS statistics computed from AMV in the CESM-LME simulations are significantly related with the AMOC-based EWS statistics (see the response to comment 1, Fig R1-R3).

Lines 496-497: 'AMV time series which by construction only reflects internal variability and thus might well reflect AMOC pulses' – is this ever assessed/shown?

The reviewer may refer to our reply to main comment 1) and also to the previous comment. However, this sentence was removed in the re-writing process.

Line 446: 'Early warning signal for an approaching tipping point' – The paper seems to end on a Results section, with a brief climate disaster warning based on the inferences made from the results. Can the authors mention anywhere the potential weaknesses/assumptions made throughout (eg. forced signal in CMIP models, AMOC=AMV)? Or mention where we go from here as a scientific community? Off the top of my head, these results also suggest a major need to better understand and observe the actual AMOC over a long enough time period to assess the hypothesized multidecadal relationship with AMV, for example, instead of relying on model simulations.

This discussion is indeed relevant in the context of our study. In the Discussion section, we have added:

"It must also be recognized that there is still a major need for future observations of the actual AMOC strength, in order to determine the extent to which the EWS observed at the ocean surface informs its future variations other than through models and paleoclimatic data".

Line 795: Why this arbitrary cutoff at the equator? Clearly proxies in Asia can show correlations with the AMV region, so why not the tropical South Atlantic or similar regions? The rebuttal figure R3 from the authors does seem to suggest there are several

locations in the western Pacific/SPCZ, Indian ocean, and South America that would show significant relationships with AMV.

The reviewer may refer to our reply to comment 4

Lines 803-804: 'Interestingly, this decision did not majorly change the final reconstruction.' can the authors show this somewhere, present results from a statistical comparison? Or at least admit it's qualitatively similar?

This proxy record was included in the first version of the manuscript and then removed for its abnormally large weight in the reconstruction and related bias. We believe that since hundreds of proxy records were used for this study, such an extreme values is not unlikely to occur by chance. In this first version of the manuscript, results obtained for the reconstruction including this proxy record (volcanic eruptions, EWS,...) were similar to that of the result from the current reconstruction. In this part of the Methods section, we specify that the reconstructions including this record is significantly correlated with the current one but apparent differences remain ($r=0.6$, $p<0.01$, Fig R8). To illustrate the bias implied by the inclusion of the Asia.MOR1JU proxy, its timeseries is also shown in Fig. R5. Other details were also given in our response to reviewer 3 at the previous round of reviews.

Figure R8: Best AMV reconstruction obtained by not using Asia.MOR1JU (blue) and using Asia.MOR1JU (orange). Pink line is the time series of Asia.MOR1JU.

Lines 830-end: unable to review Methods due to lack of ability to read symbols, etc.

We apologize for this. The reviewer may refer to our reply to main comment 5.

Lines 1023-1029: What data do boxplots show in the figures? I could not determine this from the figure caption, or from this Methods section.

The figure caption has been rewritten to explain this clearer.

Figure 3: this is a great test of the reconstruction, but seems to be a Methods/Supplement figure for a Nature Communications paper

Following the reviewer's advice, we decided to move this figure to the Supplemental. However, we believe that the two validations we made (this one and the pseudo-proxy) are essential parts of our study and that they distinguish our reconstruction of the AMV from those proposed by previous studies based on regression methods. For these reasons, we have Supplementary Notes 3 and 4 that discuss these two important validations. In the main text, these two validations are now briefly presented, with very limited technical details. This figure is now Supplementary Fig. 8.

Figure 4: how are the boxplots made? I understand the statistics they show, but I don't understand the individual data points that go into the boxplots- what is the distribution showing? Is the time series divided up within each reconstruction in the CESM runs 1-13? Or what data go into the boxplots (other than the model-derived AMV and the pseudoproxy AMV in model space, I get this)

The whisker and box plots show the CE score distribution for all the different reconstructions that have been performed by considering all the combinations of three different methodological choices: the reconstruction method, the temporal window over which it is applied and the AMV definition that was considered for the pseudo-proxy experiment.

Also, again, this Figure seems like a Supplemental/Methods type of analysis, not the main result

This figure has been moved to the supplementary but keeps being discussed in the manuscript, but to a lesser extent than the former version (see our response to the comment on Fig. 3)

Figure 5: I don't understand what the purple lines mean, or why they are lines and not just points (?) is there a connection between the CESM LME ensemble members the reader is supposed to follow, ensemble 1-13?

Thanks for spotting this. The lines are indeed not needed since each simulation member is considered independently from the others. Only points are shown in the new version of this figure (same as for former Fig. 4) to avoid any misunderstanding. Note that these figures (ex Fig. 4 and 5) are now Supplementary Fig. 9 and 10.

And again, see my comment from Figure 4- what data go into the boxplots?

We refer the reviewer to our response to the previous comment.

Figure 6: 'Composite series are performed for 31 years, with the 11th year being the year of the eruptions. Each individual response is centered to its values 10 years before the eruption (from N-10 to N-1)' – this explanation was confusing to me. Isn't year 0 the year of the eruption, and year negative 10 the 10th or 11th year before the eruption?

'centered to its value'- does this mean it's normalized or the mean is removed from this time period?

We have completely rephrased the figure caption to explain more clearly the procedure.

References:

Ba, J., Keenlyside, N. S., Latif, M., Park, W., Ding, H., Lohmann, K., ... & Volodin, E. A multi-model comparison of Atlantic multidecadal variability. *Climate dynamics* **43(9)**, 2333-2348 (2014).

Boers, N. Observation-based early-warning signals for a collapse of the Atlantic Meridional Overturning Circulation. *Nature Clim. Change* **11(8)**, 680-688 (2021).

Caesar, L., Rahmstorf, S., Robinson, A., Feulner, G., & Saba, V. Observed fingerprint of a weakening Atlantic ocean overturning circulation. *Nature* **556**, 191-195 (2018).

Frankignoul, C., Gastineau, G. & Kwon, Y.-O. Estimation of the SST Response to Anthropogenic and External Forcing and Its Impact on the Atlantic Multidecadal Oscillation and the Pacific Decadal Oscillation. *J. Clim.* **30(24)**, 9871-9895 (2017).

Gastineau, G., L'hévéder, B., Codron, F., & Frankignoul, C. Mechanisms determining the winter atmospheric response to the Atlantic overturning circulation. *Journal of Climate* **29(10)**, 3767-3785 (2016).

Harris, I., Osborn, T. J., Jones, P. & Lister, D. Version 4 of the CRU TS monthly high-resolution gridded multivariate climate dataset. *Sci. Data.* **7(1)**, 109 (2020).

Jackson, L. C. & Wood, R. A. Hysteresis and resilience of the AMOC in an eddy-permitting GCM. *Geophys. Res. Lett.* **45(16)**, 8547-8556 (2018).

Kim, W. M., Yeager, S., Chang, P., & Danabasoglu, G. Low-frequency North Atlantic climate variability in the Community Earth System Model large ensemble. *Journal of Climate* **31(2)**, 787-813 (2018).

Knight, J. R., Allan, R. J., Folland, C. K. Vellinga, M. & Mann, M. E. A signature of persistent natural thermohaline circulation cycles in observed climate. *Geophys. Res. Lett.* **32(20)**, L20708 (2005).

McCarthy, G. D., Haigh, I. D., Hirshi, J. J.-M., Grist, J. P., Smeed, D. A. Ocean impact on decadal Atlantic climate variability revealed by sea-level observations. *Nature* **521**, 508-512 (2015).

Menary, M. B., Robson, J., Allan, R. P., Both, B. B. B., Cassous, C., Gastineau, G., Gregory, J., Hodson, D., Jones, C., Mignot, J., Ringer, M., Sutton, R., Wilcox, L and Zhang, R. Aerosol-forced AMOC changes in CMIP6 historical simulations. *Geophys. Res. Lett.* **47(14)**, e2020GL088166 (2020).

Michel, S., Swingedouw, D., Ortega, P., Khodri, M., Mignot, J. & Chavent, M. Reconstructing climatic modes of variability from proxy records using ClimIndRec version 1.0. *Geosci. Mod. Dev.* **13**, 841-858 (2020).

Muir, L. C. & Fedorov, A. V. How the AMOC affects ocean temperatures on decadal to centennial timescales: the North Atlantic versus an interhemispheric seesaw. *Clim. Dynam.* **45(1-2)**, 151-160 (2015).

Persechino, A., Mignot, J., Swingedouw, D., Labetoulle, S. & Guilyardi, E. Decadal predictability of the Atlantic Meridional Overturning Circulation and climate in the IPSL-CM5A-LR model. *Clim. Dyn.* **40**, 2359-2380 (2013).

Yan., X., Zhang., R. & Knutson, T. Underestimated AMOC variability and implications for AMV and predictability in CMIP models. *Geophys. Res. Lett.* **45**, 4319-4328 (2018).

Reviewers' Comments:

Reviewer #4:

Remarks to the Author:

Re-review of

A 1150-year-long AMV reconstruction suggests an early warning for a North Atlantic climate tipping point

Overview:

The authors have generally done a good job addressing my concerns- I thank them for their time and efforts.

I have two main comments remaining:

1. The one major confusing/contradictory scientific piece of the paper for me is that the authors have done a good job justifying/showing how they removed the forced signal in their AMV reconstruction. However, the authors then seem to use the 'unforced' AMV/AMOC reconstruction to suggest that there is a forced tipping point in AMOC. How is it possible for the authors to recover a forced tipping point if the forced signal for the AMV/AMOC has been effectively removed in the reconstruction process? This seems inherently contradictory, unless I am missing an explanation of how this is possible somewhere in the text.

2. The authors generally fixed most of my wording concerns, but there is still quite a bit of confusing/unclear wording, which I have tried to note and fix in my line comments below.

Specific line comments:

Line 20: 'strong impacts in' – suggest 'on' in place of 'in'

Line 38: 'on a broader scale, it' – what is 'it' (AMV?)? - please be more specific.

Line 47: 'Seminal' – word seems extraneous- who not just 'previous work has considered'?

Line 52: I don't think the acronym 'NASST' has been defined yet

Lines 64-66: thanks to the authors for including this/highlighting the lit on AMV vs AMOC in climate models

Line 83: 'to ease its interpretation'- suggest removing- not sure what this means

Lines 93-95: can the authors better introduce or split up the long list of previous studies in this paragraph? Some of the papers are about spectral peaks, others about presence of variability, so what is the uniting theme in this paragraph?

It seems that it's about spectra, and if the variability is local vs global, and if the variability is forced or unforced (in models and in paleo data?) – could be split into several paragraphs or better introduced

Lines 103-105: 'However, while...': this concept seems to drop in suddenly- Can the authors perhaps create a new paragraph and transition- I was confused about if this is a continuation of the spectra discussion- the flow of this paragraph needs better organization/clarity.

Line 119: how is there a 'fate' of AMV? Do the authors mean 'future trajectory'? Or do the authors mean that the future change in magnitude of trends or variability is uncertain? I understand how the 'fate' of AMOC could be in question, but what is the 'fate' of an internal mode of variability (AMV), and

has the 'fate' of the AMV been questioned?

Lines 127-128: 'variations in observations and paleoclimate data...' not sure what this means- as it related to AMOC reconstructions, or just obs vs paleo can disagree? on what?

Line 140: 'reaching' – suggest 'extending'

Line 146: 'Precipitation fingerprints' - I suggest 'an analysis of precipitation suggests a fingerprint of AMV impact on precipitation' or something along those lines- I was confused by what a 'precipitation fingerprint' is

Line 172: 'large and recent' – subjective in my opinion- can authors just remove these descriptors?

Line 180: 'robustly validated'- again, subjective and extraneous in my opinion- why not just 'validated'?

Line 198: '850CE to the present'- why limit reconstruction to 850CE? Authors explain why 1870 for instrumental data, please explain why for paleo recon cutoff date

Line 200: 'which is realized separately' – I don't know what this means.

Line 286: 'fairly distributed' – what does this mean? Do the authors mean to say 'approximately equally distributed'?

Line 358-359: 'Therefore one can question the exact ability...'- this is confusing wording- 'ability' or 'evidence for'?

Lines 490-502: 'From this AMV construction, we also found...'- Please see my main comment (1) above: the one major confusing/contradictory scientific piece of the paper for me is that the authors have done a great job justifying removal of the forced signal from their reconstruction. However, the authors then seem to use the 'unforced' AMV/AMOC reconstruction to suggest that there is a forced tipping point. How is it possible for the authors to recover a forced tipping point if the forced signal for the AMV/AMOC has been effectively removed in the reconstruction process? This seems inherently contradictory, unless I am missing an explanation of how this is possible somewhere in the text.

Line 518: 'Whatsoever' – do the authors mean 'however'?

Figure 1 caption: 'relating Annual...'- here and elsewhere in figure caption(s), I suggest 'showing' in place of 'relating' – perhaps this is a personal wording preference, but it seems more clear to me.

Figure 2 caption: 'Red line:...' – Please be more consistent in your wording- it is confusing right now. I suggest something like: 'Red line: 10 year smoothed, black line: annually resolved' (confusing/inconsistent how info is presented now- one color presented in parentheses, the other in the sentence)

Response to reviewer #4

Reviewer's comments are written with normal font. **The response from the authors is written in bold font.**

1. The one major confusing/contradictory scientific piece of the paper for me is that the authors have done a good job justifying/showing how they removed the forced signal in their AMV reconstruction. However, the authors then seem to use the 'unforced' AMV/AMOC reconstruction to suggest that there is a forced tipping point in AMOC. How is it possible for the authors to recover a forced tipping point if the forced signal for the AMV/AMOC has been effectively removed in the reconstruction process? This seems inherently contradictory, unless I am missing an explanation of how this is possible somewhere in the text.

We thank the reviewer for bringing this important point to our attention, which was possibly unclear when the manuscript was written. We believe this misunderstanding stems from the word "unforced," which can have several meanings.

In this paper, the term "unforced" refers to a climate index that has had the direct effect of external radiative forcing on SST removed. By doing so for the AMV, we remove the part of the signal that is driven by external radiative forcing (e.g., anthropogenic and volcanic aerosols, greenhouse gases), but not the part that is driven by ocean dynamics, most notably the intrinsic ocean variability of the North Atlantic system.

Indeed, we argued that if this external signal is not removed, this can be a serious issue in terms of process understanding, because we may mix large-scale signals with different patterns (e.g. global for greenhouse forcing) and drivers (radiative vs. oceanic). One of the key aspects of the manuscript is to focus on the part of the North Atlantic SST that is primarily driven by oceanic processes and air-sea interactions, known in the literature as the AMV, where the direct impact of external radiative forcing is removed from the SST to obtain what we previously referred to as the "unforced" signal. However, this does not mean that the oceanic processes driving the AMV have been completely cleaned up of all external forcings. Other indirect and delayed effects of external forcings on the AMV, such as changes in AMOC heat transport (e.g., freshwater input), is still captured by this index. This, in particular, allows for the observation of changes in intrinsic variability caused by minor changes in the mean state of the analyzed dynamical system as a result of external forcing perturbation (e.g. Lenton 2011, its Fig. 2).

Indeed, early warning signals, according to tipping point theory, are looking at the signal of a system in which intrinsic variability is changing in response to a perturbation that is not directly affecting the analyzed variable, but is changing the mean state of the system and thus its variability properties, which are reflected in the analyzed variable. Indeed, early warning theory seeks a dynamical response in terms of intrinsic variability to changes in mean state caused by a perturbation, which leads to an increase in the system's memory

when approaching a tipping point (cf. Lenton 2011). In this regard, the use of an AMV index that excludes radiative forcing allows us to better capture the fingerprints of changes in variability properties of the North Atlantic dynamical system, and more specifically, the way the intrinsic variability of this system evolves in response to external forcing.

We have extensively clarified our approach in the new version of the manuscript to avoid any confusion for readers.

2. The authors generally fixed most of my wording concerns, but there is still quite a bit of confusing/unclear wording, which I have tried to note and fix in my line comments below.

We thoroughly revised the text in order to improve the manuscript's readability.

Line 20: 'strong impacts in' – suggest 'on' in place of 'in'

Done

Line 38: 'on a broader scale, it' – what is 'it' (AMV)? - please be more specific.

Done

Line 47: 'Seminal' – word seems extraneous- who not just 'previous work has considered'?

This word was intended to insist that this view was almost exclusively considered in the first publications of the AMV (2000's), followed by the controversy concerning the effect of aerosols (early 2010's). We replaced it with "Early studies".

Line 52: I don't think the acronym 'NASST' has been defined yet

Good catch. It has been corrected.

Lines 64-66: thanks to the authors for including this/highlighting the lit on AMV vs AMOC in climate models

Line 83: 'to ease its interpretation'- suggest removing- not sure what this means

This was removed.

Lines 93-95: can the authors better introduce or split up the long list of previous studies in this paragraph? Some of the papers are about spectral peaks, others about presence of variability, so what is the uniting theme in this paragraph?

It seems that it's about spectra, and if the variability is local vs global, and if the variability is forced or unforced (in models and in paleo data?) – could be split into several paragraphs or better introduced

Lines 103-105: 'However, while...': this concept seems to drop in suddenly- Can the authors perhaps create a new paragraph and transition- I was confused about if this is a continuation of the spectra discussion- the flow of this paragraph needs better organization/clarity.

We have fully rewritten those paragraphs to better highlight the unity of the different paragraphs. We hope the reviewer will find them better organized by topic.

Line 119: how is there a 'fate' of AMV? Do the authors mean 'future trajectory'? Or do the authors mean that the future change in magnitude of trends or variability is uncertain? I understand how the 'fate' of AMOC could be in question, but what is the 'fate' of an internal mode of variability (AMV), and has the 'fate' of the AMV been questioned?

Indeed, "its" was associated with the AMOC but appears not clear. It was replaced by "the AMOC fate".

Lines 127-128: 'variations in observations and paleoclimate data...' not sure what this means- as it related to AMOC reconstructions, or just obs vs paleo can disagree? on what?

This sentence was re-written: "Other recent studies, however, have revealed that the strength of ocean currents varies widely between regions, ocean depths, and time^{36,37} casting doubt on the hypothesis of a recent major change in the AMOC based on a relatively small number of ocean cores³¹⁻³³"

Line 140: 'reaching' – suggest 'extending'

Done

Line 146: 'Precipitation fingerprints' - I suggest 'an analysis of precipitation suggests a fingerprint of AMV impact on precipitation' or something along those lines- I was confused by what a 'precipitation fingerprint' is

This sentence was rephrased: "The AMV fingerprints on precipitation show that AMV positive phases are [...]"

Line 172: 'large and recent' – subjective in my opinion- can authors just remove these descriptors?

Done

Line 180: 'robustly validated'- again, subjective and extraneous in my opinion- why not just 'validated'?

This was replaced as suggested by the reviewer.

Line 198: '850CE to the present'- why limit reconstruction to 850CE? Authors explain why 1870 for instrumental data, please explain why for paleo recon cutoff date

In the former version, we specified that 850 C.E. was chosen so that the reconstruction covers the same time frame as CESM-LME simulations and can be tested in a pseudo-proxy experiment framework, but this was removed during the revision process. This information was added back in the new version of the manuscript: "The year 850 C.E. was chosen since it is the starting point for the last millennium climate simulations. Such simulations will be used later to further evaluate and validate our reconstruction in a pseudo-proxy framework."

Line 200: 'which is realized separately' – I don't know what this means.

We replaced this sentence by "[...], which is performed independently for each of the three AMV definitions used".

Line 286: 'fairly distributed' – what does this mean? Do the authors mean to say 'approximately equally distributed'?

This was replaced by “similarly redistributed”.

Line 358-359: ‘Therefore one can question the exact ability...’- this is confusing wording- ‘ability’ or ‘evidence for’?

This was replaced by “Therefore, the volcanic forcing is not found to act as a pacemaker for the AMV, at least for the 10 largest volcanic eruptions of the last millennium.”

Lines 490-502: ‘From this AMV construction, we also found...’- Please see my main comment (1) above: the one major confusing/contradictory scientific piece of the paper for me is that the authors have done a great job justifying removal of the forced signal from their reconstruction. However, the authors then seem to use the ‘unforced’ AMV/AMOC reconstruction to suggest that there is a forced tipping point. How is it possible for the authors to recover a forced tipping point if the forced signal for the AMV/AMOC has been effectively removed in the reconstruction process? This seems inherently contradictory, unless I am missing an explanation of how this is possible somewhere in the text.

The reviewer may refer to our response to comment 1.

Line 518: ‘Whatsoever’ – do the authors mean ‘however’?

We have replaced it with “however”.

Figure 1 caption: ‘relating Annual...’- here and elsewhere in figure caption(s), I suggest ‘showing’ in place of ‘relating’ – perhaps this is a personal wording preference, but it seems more clear to me.

We have made the suggested modification.

Figure 2 caption: ‘Red line:...’ – Please be more consistent in your wording- it is confusing right now. I suggest something like: ‘Red line: 10 year smoothed, black line: annually resolved’ (confusing/inconsistent how info is presented now- one color presented in parentheses, the other in the sentence)

We have homogenized the figure’s legend as suggested by the reviewer.

Reference:

Lenton, T. M. Early warning of climate tipping point. *Nature Clim. Change* **1**, 201-208 (2011).

Reviewers' Comments:

Reviewer #4:

Remarks to the Author:

Re-review of:

A 1150-year-long AMV reconstruction suggests an early warning for a North Atlantic climate tipping point

Summary:

The authors have addressed all of my concerns, and the paper is significantly easier to read/follow than both of the previous versions I reviewed. I found a few minor wording issues, detailed below, but overall, I have no remaining major concerns with the manuscript. I focused my comments on the main text as this was where most of my previous concerns were concentrated.

Line comments:

Abstract:

'Atlantic multidecadal variability is a large-scale climate phenomenon in North Atlantic sea surface temperatures with strong impacts on human societies and ecosystems worldwide.'

I suggest a less convoluted way of describing the AMV- see for example the NCAR Climate Data Guide description: 'The Atlantic Multi-decadal Oscillation (AMO) has been identified as a coherent mode of natural variability occurring in the North Atlantic Ocean ' -I wasn't sure for example: is the phenomenon large-scale? Or are the impacts large scale? Etc.

Page 2: 'dampening' implies getting wetter- do the authors intend to use 'damping'?

Page 5: 'of the latter study' - suggest clarifying as I wasn't sure what the latter study is- is it the next one mentioned, or the previous study from the previous sentence? Or the second study in the paragraph?

Page 12: 'or simply indicate'- suggest 'or could simply indicate'

Page 14: 'separated from instant radiative'- suggest 'was separated from' or similar wording

Page 15: 'it plays a direct role'- in my opinion, statistic doesn't indicate this is incredibly robust (pval ~0.2?), so I suggest less strong wording such as 'it could play a direct role'

Figure 6 and others: font size is so small in text within subplots that I can barely read it after zooming in- please check font size across figures for readability

Response to Reviewer #4

Reviewer's comment are written in normal plain text whereas the authors' responses are **written in bold**.

Re-review of:

A 1150-year-long AMV reconstruction suggests an early warning for a North Atlantic climate tipping point

Summary:

The authors have addressed all of my concerns, and the paper is significantly easier to read/follow than both of the previous versions I reviewed. I found a few minor wording issues, detailed below, but overall, I have no remaining major concerns with the manuscript. I focused my comments on the main text as this was where most of my previous concerns were concentrated.

Line comments:

Abstract:

'Atlantic multidecadal variability is a large-scale climate phenomenon in North Atlantic sea surface temperatures with strong impacts on human societies and ecosystems worldwide.'

I suggest a less convoluted way of describing the AMV- see for example the NCAR Climate Data Guide description: 'The Atlantic Multi-decadal Oscillation (AMO) has been identified as a coherent mode of natural variability occurring in the North Atlantic Ocean' –I wasn't sure for example: is the phenomenon large-scale? Or are the impacts large scale? Etc.

We have made the modification suggested by the reviewer.

Page 2: 'dampening' implies getting wetter- do the authors intend to use 'damping'?
We have replaced this term by "damping".

Page 5: 'of the latter study' – suggest clarifying as I wasn't sure what the latter study is- is it the next one mentioned, or the previous study from the previous sentence? Or the second study in the paragraph?

This point is now made clearer using "the aforementioned study".

Page 12: 'or simply indicate'- suggest 'or could simply indicate'

This modification has been made.

Page 14: 'separated from instant radiative'- suggest 'was separated from' or similar wording

This modification has been made.

Page 15: 'it plays a direct role'- in my opinion, statistic doesn't indicate this is incredibly robust (pval ~0.2?), so I suggest less strong wording such as 'it could play a direct role'

We have made the modification suggested by the reviewer.

Figure 6 and others: font size is so small in text within subplots that I can barely read it after zooming in- please check font size across figures for readability

Font size of all figures has been increased significantly.